# Rationalizing kinetic behaviors of isolated boron sites catalyzed oxidative dehydrogenation of propane

Hao Tian [1,2], Wenying Li[3], Linhai He[4], Yunzhu Zhong[1,2], Shutao Xu [4], Hai Xiao [3] & Bingjun Xu [1,2] ✉

Boron-based catalysts exhibit high alkene selectivity in oxidative dehydrogenation of propane (ODHP) but the mechanistic understanding remains incomplete. In this work, we show that the hydroxylation of framework boron species via steaming not only enhances the ODHP rate on both B-MFI and B-BEA, but also impacts the kinetics of the reaction. The altered activity, propane reaction order and the activation energy could be attributed to the hydrolysis of framework $[B(OSi\equiv)_3]$ unit to $[B(OSi\equiv)_{3-x}(OH\cdots O(H)Si\equiv)_x]$ ($x = 1, 2$, "$\cdots$" represents hydrogen bonding). DFT calculations confirm that hydroxylated framework boron sites could stabilize radical species, e.g., hydroperoxyl radical, further facilitating the gas-phase radical mechanism. Variations in the contributions from gas-phase radical mechanisms in ODHP lead to the linear correlation between activation enthalpy and entropy on borosilicate zeolites. Insights gained in this work offer a general mechanistic framework to rationalize the kinetic behavior of the ODHP on boron-based catalysts.

Propylene ($C_3H_6$) is a fundamental building block of the chemical industry to produce bulk chemicals such as polypropylene, acrylonitrile, propylene oxide and acrylic acid[1]. Non-oxidative propane ($C_3H_8$) dehydrogenation (PDH) has been commercialized to meet the growing demand of $C_3H_6$, but the efficiency of PDH is limited by the unfavorable thermodynamics and coking-induced rapid catalyst deactivation[2]. Oxidative dehydrogenation of propane (ODHP), which introduces molecular oxygen ($O_2$) to facilitate the C−H activation of propane, has the potential to overcome the challenges faced by PDH[3,4]. However, conventional ODHP catalysts, such as vanadium-based catalysts, exhibit poor alkene selectivity, which is caused by the deep oxidation of $C_3H_6$ to form carbon monoxide (CO) and carbon dioxide ($CO_2$)[5]. The development of catalysts with high $C_3H_6$ yield and long-term stability is critical to make ODHP commercially viable.

ODHP activity of supported boron oxide ($B_2O_3$) catalysts has been recognized since the late 1980s, and the first application of hexagonal boron nitride (h-BN) after oxyfunctionalization was reported in 2016[6–8]. Distinct from vanadium-based catalysts, boron-based catalysts are able to suppress further oxidation of alkenes in the presence of $O_2$[8,9]. A second-order dependence on the partial pressure of $C_3H_8$ ($p_{C3H8}$) and the apparent activation energy ($E_{app}$) higher than 150 kJ·mol$^{-1}$ were observed on most boron-based catalysts (Table S1). The supra-linear reaction order of $C_3H_8$ cannot be rationalized by classical surface-mediated mechanisms in heterogeneous catalysis, which led to much research effort in understanding the catalytic mechanism. Hermans and co-workers found that the conversion rate of $C_3H_8$ was proportional to the volume of catalyst bed, rather than the catalyst loading, in the h-BN catalyzed ODHP, implying that a

[1]College of Chemistry and Molecular Engineering, Peking University, Beijing 100871, China. [2]Beijing National Laboratory for Molecular Sciences, Beijing 100871, China. [3]Department of Chemistry and Key Laboratory of Organic Optoelectronics & Molecular Engineering of Ministry of Education, Tsinghua University, Beijing 100084, China. [4]National Engineering Laboratory for Methanol to Olefins, Dalian National Laboratory for Clean Energy, iChEM (Collaborative Innovation Center of Chemistry for Energy Materials), Dalian Institute of Chemical Physics, Chinese Academy of Sciences, Dalian 116023, China. ✉e-mail: b_xu@pku.edu.cn

surface-mediated pathway is unlikely to be able to account for the majority of observed activity[10]. They proposed a combined surface and gas-phase radical reaction mechanism capable of rationalizing the product distribution of ODHP observed on h-BN[11]. Our recent kinetic study on oxidative co-dehydrogenation of ethane ($C_2H_6$) and $C_3H_8$ over h-BN found that the apparent second-order dependence on the partial pressure of alkane in ODHP could be rationalized by two roles played by the alkane, i.e., radical generator and reactant[12]. The high propylene selectivity of bulk boron-based catalysts in ODHP is likely due to the involvement of gas-phase radicals in activating propane, though the exact mechanism for suppressing the further conversion of propylene remains unclear[13–15]. The study using the magic angle spinning (MAS) $^{11}B$ solid-state nuclear magnetic resonance (NMR) spectroscopy revealed that oxidized boron species, denoted as $B(OH)_xO_{3-x}$ (where $x = 0–3$), were gradually formed on h-BN in ODHP[16,17]. Similar oxidized boron species could also be formed on other boron-containing materials under ODHP conditions, and are likely responsible for the catalytic activity[18].

The identities of active structure of isolated boron sites and reaction mechanism are less clear. Hermans and coworkers synthesized boron-substituted MWW (B-MWW) zeolites with negligible ODHP activity[19]. $^{11}B$ NMR results showed that most framework boron atoms were in the trigonal coordination environment, which led to the conclusion of isolated boron sites being inactive. Meanwhile, B-MWW prepared by Lu and coworkers showed quite similar overall and deconvoluted NMR features and exhibited high ODHP activity[20]. Xiao and coworkers synthesized boron-substituted MFI- and BEA-type (B-MFI and B-BEA) zeolites with most framework boron atoms located in the tetrahedral coordination environment both in as-synthesized and spent samples[21]. These B-MFI and B-BEA catalysts showed remarkable ODHP activities and durability. A first-order kinetics with respect to $p_{C3H8}$ was determined on both borosilicate zeolites. Combined with computational results, a surface-mediated mechanism was proposed on trigonally coordinated framework boron atoms in ODHP. Qiu et al. assigned aggregated boron species in incompletely crystallized B-MFI as the most active structure[22,23]. These inconsistent literature claims highlight the need for further mechanistic studies.

Determination of kinetic parameters, i.e., the reaction order of $C_3H_8$ and the apparent activation energy ($E_{app}$), proves to be informative in mechanistic studies of boron-catalyzed ODHP. The likely involvement of gas-phase radicals, as well as the high reaction temperature, in ODHP makes the detection of surface intermediates challenging and less relevant. In contrast, $C_3H_8$ reaction order could serve as a diagnostic variable to identify the dominant type of mechanism at play. A first-order kinetics for $C_3H_8$ indicates a surface-mediated mechanism[24–26], which is typical on vanadium-based catalysts. Most bulk boron-based catalysts exhibit a second-order reaction for $C_3H_8$, indicating the involvement of a gas-phase radical mechanism[11,12]. Our recent work showed that spatial confinement of $B_2O_3$ nanoparticles in mesoporous support could lead to $C_3H_8$ reaction orders up to 3[27], which was attributed to the introduction of a branching radical pathway.

In this study, we investigated the correlation between the boron coordination environment and the ODHP kinetics. Fresh B-MFI with majority of boron species incorporated in the framework exhibits a first-order dependence on $p_{C3H8}$ and $E_{app}$ ranging from 100 to 120 kJ·$mol^{-1}$. Steaming treatment gradually hydrolyzes the B–O–Si bonds in B-MFI, which proceeds from $[B(OSi\equiv)_3]$ to hydroxyl group substituted $[B(OSi\equiv)_{3-x}(OH\cdots O(H)Si\equiv)_x]$ ($x = 1, 2$), and enhances the ODHP activity by up to a factor of 9 on B-MFI. Computational results show that the hydroxylation of the isolated boron sites could stabilize potential gas-phase radical species derived from $O_2$, facilitating the initiation of the radical chain mechanism. Similar observations were made on B-BEA. Degree of hydroxylation on framework boron

species is proposed to be the key in understanding the kinetic behaviors of boron-containing zeolites after different pretreatment conditions and in different ranges of $p_{C3H8}$.

## Results

### Boron coordination and ODHP performance of the fresh B-MFI Zeolite

B-MFI was synthesized via a solvent-free crystallization method reported by Zhou et al.[21] as conventional hydrothermal method was prone to forming non-framework boron species[28,29]. The as-prepared sample is referred to as fresh B-MFI below. Fresh B-MFI consists of sphere-like particles with an average diameter of ~1 µm (Fig. S1). XRD patterns of fresh B-MFI, H-ZSM-5 and Silicalite-1 exhibit diffraction peaks characteristic of the MFI-type zeolites with no detectable phase impurities. (Fig. S2) Lattice parameters and unit cell volumes of the MFI-type zeolites were calculated based on the peak positions in the XRD patterns. (Table S2) Compared to H-ZSM-5 and Silicalite-1, the unit cell volume of B-MFI is slightly contracted, indicating that the boron atoms are successfully incorporated into the framework of zeolite[30]. Micropore volume of fresh B-MFI, H-ZSM-5 and Silicalite-1 are in line with typical MFI-type zeolites[31]. (Fig. S3 and Table S3)

To understand the coordination environment of the boron species, $^{11}B$ and $^{29}Si$ NMR spectra of fresh B-MFI were collected. (Figs. S4 and S5) $^{11}B$ chemical shift of fresh B-MFI shows a strong peak at −3.2 ppm and a broad peak at 5.9 ppm. The former has been assigned to the tetra-coordinated boron species in zeolite frameworks (denoted as B[4]-Fr)[32]. The band at −3.2 ppm exhibits a Lorentzian lineshape, which is in line with B[4]-Fr[33]. Distinct from tetrahedrally coordinated boron species, the strong quadrupolar interaction of trigonally coordinated boron leads to a peak shift and anisotropic line broadening of tri-coordinated boron species[34]. Combing the chemical shift and the lineshape, the peak at 5.9 ppm could be attributed to tri-coordinated boron species in the zeolite framework (denoted as B[3]-Fr)[35]. Non-framework tri-coordinated boron species (denoted as B[3]-Nf), for example $H_3BO_3$ ($\delta_{iso} = 18.0–18.5$ ppm)[36,37] and $B_2O_3$ ($\delta_{iso} = 14.6$ ppm)[38], are absent in fresh B-MFI, indicating that most boron atoms are incorporated in the framework and isolated by neighboring −O−Si−O− units. $^{29}Si$ NMR spectrum of fresh B-MFI exhibits a strong peak at −113 ppm and a shoulder peak at −103 ppm, corresponding to $[Si(OSi\equiv)_4]$ and $[Si(OSi\equiv)_3OH]$ in the zeolite framework, respectively[39]. (Fig. S5) According to the peak area in Fig. S4, B[4]-Fr is estimated to account for 92% of all boron atoms at room temperature in fresh B-MFI, which is consistent with a previous study[21].

Although most framework boron atoms are in the form of B[4]-Fr at room temperature, in-situ IR spectroscopy shows that B[3]-Fr is the dominant species at reaction temperature. To study the boron coordination under the ODHP conditions, transmission IR spectra of dehydrated fresh B-MFI at different temperatures were collected. (Fig. S6) The peak around 3720 $cm^{-1}$ is attributed to silanol groups (Si−OH) on zeolite and the peak at 3520 $cm^{-1}$ is assigned to the Brønsted acid site in borosilicate zeolite[39,40]. (Fig. 1) The band at 910 $cm^{-1}$ is attributed to the symmetric stretching of B[4]-Fr while the band located in the 1400 to 1360 $cm^{-1}$ range corresponds to the B−O asymmetric stretching vibration of B[3]-Fr[34,41]. (Fig. 1) As temperature rises, the intensity of IR peaks at 3520 and 910 $cm^{-1}$ decrease concomitantly while the peak at 1380 $cm^{-1}$ strengthens, indicating that B[4]-Fr is transformed to B[3]-Fr at temperatures relevant in ODHP. Together with the absence of IR band for B−OH at 3680 $cm^{-1}$ [37,42], it could be inferred that $[B(OSi\equiv)_3]$ unit is the dominant boron specie of fresh B-MFI at the ODHP temperature[21,29,41].

Fresh B-MFI exhibits distinct catalytic behavior from most bulk boron-based catalysts in ODHP (Fig. 2a, b). To verify the initial ODHP activity prior to activation is attributed to framework boron species, we conducted two control experiments (Table S4 and Fig. S7).

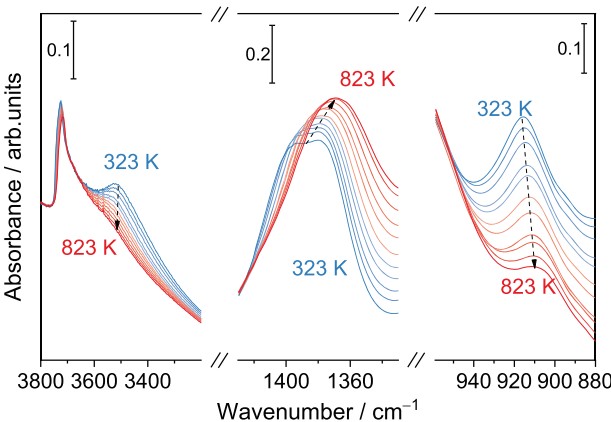

**Fig. 1 | Influence of temperature on boron coordination.** Transmission IR bands of OH groups (left part), trigonal framework boron species (middle part) and tetrahedral framework boron species (right part) on fresh B-MFI at different temperatures.

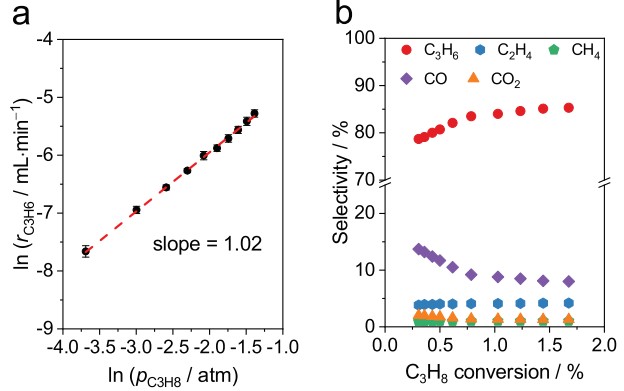

**Fig. 2 | ODHP performance on fresh B-MFI. a** The reaction order of $r_{C3H6}$ with respect to $p_{C3H8}$. Reaction conditions: 773 K, $p_{O2} = 0.125$ atm with balancing $N_2$. Error bars show represent standard deviation of ln ($r_{C3H6}$) in three replicate data. **b** Product selectivity as a function of $C_3H_8$ conversion. Reaction conditions: Reaction conditions: 773 K, $p_{C3H8} = 0.25$ atm, $p_{O2} = 0.125$ atm with balancing $N_2$, space velocity from 1500 to 24000 $L_{C3H8} \cdot kg_{cat}^{-1} \cdot h^{-1}$.

Silicalite-1 showed no ODHP selectivity to $C_3H_6$ (Table S4), and $C_3H_6$ formation rate of fresh B-MFI in ODHP exhibited a positive correlation to boron content in B-MFI (Fig. S7). The control experiments confirmed that the framework boron species are likely to be the active sites responsible for the initial ODHP activity in fresh B-MFI. The formation rate of $C_3H_6$ ($r_{C3H6}$) over fresh B-MFI exhibits a first-order dependence on $p_{C3H8}$. (Fig. 2a) This result is different from the supralinear order observed over most bulk boron-based catalysts, but in line with a recent report on borosilicates[21]. (Table S1) Product distribution in ODHP on fresh B-MFI as a function of the $C_3H_8$ conversion is shown in Fig. 2b. The propylene selectivity increases almost linearly with the propane conversion when the conversion is below 1%, and then gradually levels off at conversion above 1.5% at 85%. While ethylene ($C_2H_4$) is the main by-product in ODHP on most bulk boron-based catalysts[43], CO is the main by-product on fresh B-MFI, and its selectivity increases with decreasing $C_3H_8$ conversion. The high CO selectivity in ODHP on boron-substituted zeolites was also observed by Lin and co-workers[44]. The distinct $C_3H_8$ reaction order and product distribution on fresh B-MFI indicate that ODHP on fresh B-MFI likely proceeds via a distinct pathway from that on catalysts with aggregated boron species, e.g., $B_2O_3$ and h-BN.

## The promotion of ODHP activity via steaming treatment

Steaming treatment markedly promotes the ODHP activity on fresh B-MFI. Steaming is an effective strategy to accelerate the hydrolysis of Si−O−X bonds (X represents a heteroatom) in zeolites, and thus tuning the coordination of framework heteroatoms such as boron[45,46]. The steaming of fresh B-MFI was carried out at high temperature (823 K) and low partial pressure of water ($p_{H2O} = 1.8$ kPa). $r_{C3H6}$ increases almost linearly with the duration of the steaming treatment within the first 3 h and levels off after 4 h regardless of $p_{C3H8}$ (Fig. 3a−c). The increase in the ODHP activity after steaming is more pronounced at high $p_{C3H8}$, and the apparent reaction order of $C_3H_8$ increases with the steaming duration. (Fig. S8) The specific $C_3H_6$ formation rate on fully steamed B-MFI (with steaming duration > 4 h) reaches 0.67 $mol_{C3H6} \cdot mol_B^{-1} \cdot h^{-1}$ at $p_{C3H8} = 0.25$ atm and 803 K, which is nearly one order of magnitude higher than that of fresh B-MFI. (Fig. 3d) The ODHP activity of fully steamed B-MFI could be further enhanced in ODHP, i.e., there is an induction period[47]. (Fig. S9) Steamed B-MFI after the induction period is referred to as activated B-MFI later in this work. The fact that activated B-MFI exhibits superior ODHP activity than fully steamed sample indicates that species involved in the reaction are able to induce structrual changes at boron site that are inaccessible via steaming. The steady-state $C_3H_6$ formation rates after the induction period on fresh and fully steamed B-MFI are comparable, implying that ODHP could activate fresh B-MFI without the steaming treatment. This is expected considering that ODHP produces water, so the catalyst is subject to conditions similar to the steaming treatment during reaction. The specific $C_3H_6$ formation rate on activated B-MFI sample is 0.88 $mol_{C3H6} \cdot mol_B^{-1} \cdot h^{-1}$ at $p_{C3H8} = 0.25$ atm, (Fig. 3d) which is comparable to the PDH activity on noble metals[48]. To avoid interference of the induction period in determining the kinetic parameters of ODHP, activated B-MFI was employed in the following investigations.

Activated B-MFI exhibits a higher $E_{app}$ than that on fresh B-MFI. (Fig. 3e and Table S5) At $p_{C3H8} = 0.25$ atm, $E_{app}$ of activated and fresh B-MFI were determined to be 163 and 125 kJ·mol$^{-1}$, respectively. These measured $E_{app}$ values appear counterintuitive as the more active activated B-MFI exhibits a higher $E_{app}$. The apparent pre-exponential factor ($A_{app}$) of ODHP on activated B-MFI is higher than that of fresh B-MFI by a factor of ~400. Values of $E_{app}$ and $A_{app}$ also depend on $p_{C3H8}$. When $p_{C3H8}$ is decreased from 0.25 to 0.15 atm, both $E_{app}$ and $A_{app}$ of fresh and activated B-MFI are reduced, leading to a linear correlation between $\ln(A_{app})$ and $E_{app}$ (Fig. 3e and Table S5). Literature suggest that $E_{app}$ is typically higher with mechanisms involving gas-phase radicals (>150 kJ·mol$^{-1}$) than that with surface-mediated mechanisms (< 120 kJ·mol$^{-1}$)[27]. Thus, measured $E_{app}$ values suggest that ODHP on activated B-MFI likely occurs via a radical mechanism.

$C_3H_8$ reaction order and product distribution in ODHP on activated B-MFI differ fundamentally from those on fresh B-MFI. The apparent $C_3H_8$ order on activated B-MFI increases from 1.04 to 2.05 as $p_{C3H8}$ rises beyond 0.15 atm (Fig. 3f), while no such transition is present on fresh B-MFI (Fig. 2a). A first-order kinetics for $C_3H_8$ indicates that the reaction proceeds via a surface-mediated mechanism, while a second-order suggests a gas-phase radical mechanism[11,12,24−27]. This claim is also supported by the experiments that using silicon carbide (SiC) as a diluent to differentiate surface-mediated reactions from gas-phase radical reactions[10]. The introduction of SiC does not influence ODHP activity of fresh B-MFI, but markedly increases $C_3H_6$ formation rate of activated B-MFI (Fig. S10a). Enhancements in catalytic activity induced by diluent is a characteristic of gas-phase reaction, which could be attributed to the extended residence time of gas-phase active species (Fig. S10b). Moreover, similar to bulk boron species but distinct from fresh B-MFI, $C_2H_4$, rather than CO, is the main by-product on activated B-MFI. Further, the trends of $C_3H_6$ and CO selectivity with decreasing $C_3H_8$ conversion are opposite on fresh and activated B-MFI (Figs. 2b, 3g), indictive of distinct reaction mechanisms. $C_3H_8$ reaction order

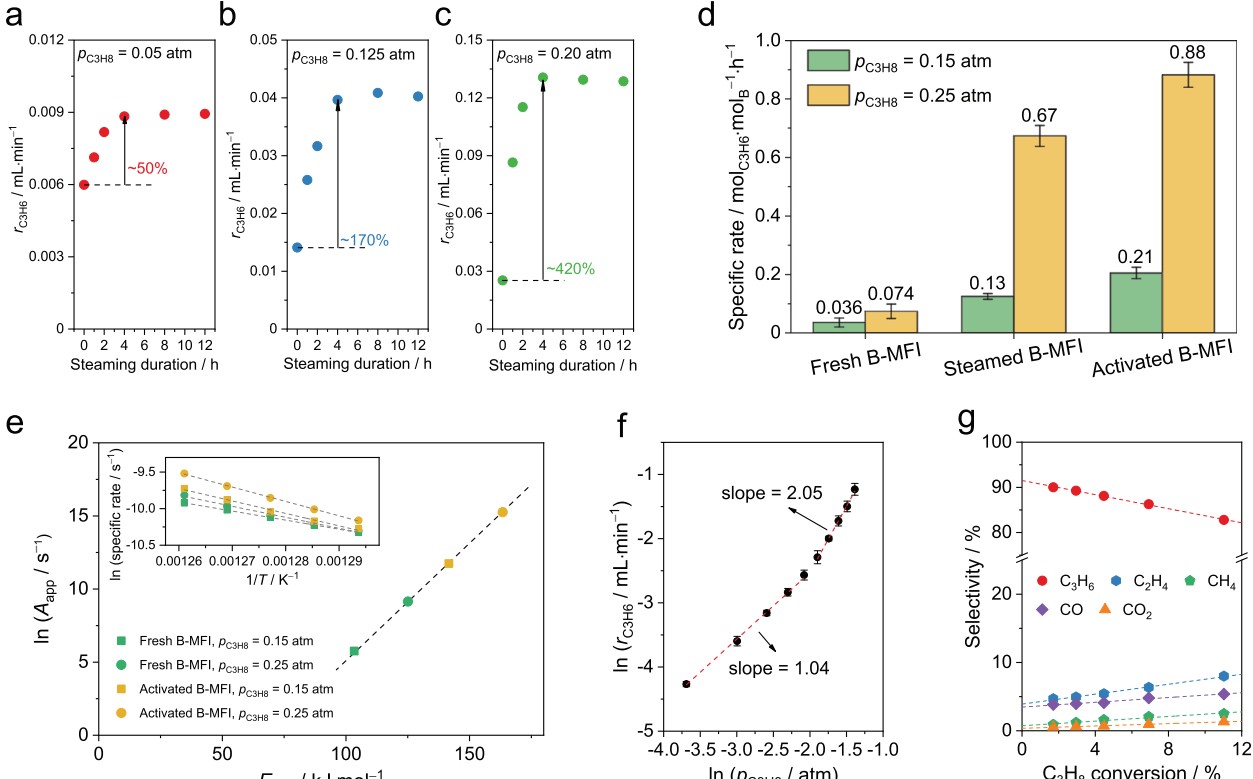

**Fig. 3 | Influence of steaming on ODHP on B-MFI. a–c** Influence of steaming duration on $r_{C3H6}$ at different $p_{C3H8}$: **a** $p_{C3H8}$ = 0.05 atm; **b** $p_{C3H8}$ = 0.125 atm; **c** $p_{C3H8}$ = 0.20 atm. **d** Specific $C_3H_6$ formation rate of different B-MFI. Error bars show represent standard deviation of specific rates in three replicate data. Reaction conditions for (**a–d**): 803 K, total gas flow = 40 mL·min$^{-1}$, 50 mg catalyst, $p_{O2}$ = 0.125 atm with balancing $N_2$. **e** Correlation between $E_{app}$ and $A_{app}$. The inset are corresponding Arrhenius plots. Reaction conditions for (**e**): 773 to 793 K, total

gas flow = 40 mL·min$^{-1}$, 50 mg catalyst, $p_{O2}$ = 0.125 atm with balancing $N_2$. **f** Reaction order of $r_{C3H6}$ with respect to $p_{C3H8}$ on activated B-MFI. Error bars show represent standard deviation of ln ($r_{C3H6}$) in three replicate data. **g** Product selectivity (% of carbon atoms in $C_3H_8$ converted to products) as a function of $C_3H_8$ conversion on activated B-MFI. Reaction conditions for (**f**, **g**): 773 K, total gas flow = 40 mL·min$^{-1}$, $p_{O2}$ = 0.125 atm with balancing $N_2$.

($p_{C3H8}$ > 0.15 atm) and product distribution on activated B-MFI are similar to those on catalysts with aggregated boron species, suggesting that the nature of active boron species on fresh B-MFI has been altered during the steaming treatment and the induction period.

**Evolution of active boron species in B-MFI**

Characterization results show that elemental composition, morphology and crystalline structure of B-MFI remain largely unchanged during the steaming treatment and in the induction period. The melting point of $B_2O_3$ (723 K) is close to ODHP temperature and the volatilization of boron species was considered as a potential cause for catalyst deactivation[43]. B/Si molar ratios of both fresh and activated B-MFI determined by ICP-AES are consistent with that employed in the synthesis, indicating negligible boron loss. (Fig. S11) SEM images, XRD pattern and $^{29}$Si NMR spectra show that activated B-MFI retains the crystalline size, morphology, crystallographic parameters and silicon coordination environment after the steaming treatment and the induction period. (Figs. S12–14 and Table S6) Characterization results indicate that the enhanced ODHP activity on activated B-MFI is unlikely caused by changes in the compositional and structural change in the framework of B-MFI.

$^{11}$B NMR results indicate that the boron coordination in fresh B-MFI is gradually hydroxylated during the steaming treatment, but most boron atoms remain in the zeolite framework in fully steamed and activated B-MFI. $^{11}$B NMR spectra of various B-MFI samples were fitted with three peaks attributed to B[4]-Fr (red), B[3]-Fr (green) and B[3]-Nf (blue) (Fig. 4a, S15 and Table S7)[29,34,49]. B[4]-Fr is the dominant boron species (88%) in fresh B-MFI, with the rest being B[3]-Fr. With

increasing steaming duration (1–4 h) at 823 K, an increasing fraction of B[4]-Fr is hydrolyzed to B[3]-Fr. B[3]-Fr bonded with hydroxyl, i.e., [B(OSi)$_x$(OH)$_{3-x}$] ($x$ = 1, 2), are stabilized by the formation of hydrogen bonding network, suppressing the conversion of B[3]-Fr with hydroxyl to B[4]-Fr via condensation[34]. After steaming for 2 h, B[3]-Nf, i.e., $B_2O_3$ and hydroxylated $B_2O_x$(OH)$_{3-x}$, appears in $^{11}$B NMR spectra due to the further hydrolysis of B−O−Si bond. The conversion from B[4]-Fr to B[3]-Fr, as well as the further hydroxylation of B[3]-Fr to B[3]-Nf, slows after 4 h of steaming. ODHP activity parallels the increase in the fraction of boron in B[3]-Fr within the first 4 h of steaming, and levels off afterwards (Fig. 3a–c). The fraction of boron species in the form of B[3]-Nf increases from 11% to 15% when B-MFI is steamed for additional 8 h after the initial 4 h, but with no appreciable change in the ODHP activity. The main difference in the distribution of various boron species between fully steamed and activated B-MFI is that B[3]-Fr accounts for a slightly higher fraction in activated B-MFI at the expense of B[4]-Fr (Fig. S15 and Table S7). The combined fraction of B[4]-Fr and B[3]-Fr in fully steamed and activated B-MFI is ≥ 85%, (Fig. S15 and Table S7) indicating that the majority of boron species are present as isolated in the zeolite framework. As discussed in Fig. 1, B[4]-Fr in $^{11}$B NMR spectra is converted into [B(OSi≡)$_3$] at high temperature. It can be inferred that B[3]-Fr species, i.e., [B(OSi≡)$_{3-x}$(OH)$_x$] characterized by the $^{11}$B NMR spectroscopy, are largely responsible for the enhanced ODHP activity during steaming treatment. (Fig. 4b) The values of fitting lines on y-axis corresponds to the intrinsic activity of [B(OSi≡)$_3$], which is the dominant boron specie on fresh B-MFI. As the duration of the steaming treatment increases, [B(OSi≡)$_3$] is gradually hydroxylated to [B(OSi≡)$_{3-x}$(OH)$_x$], leading to a linear increase in the ODHP activity.

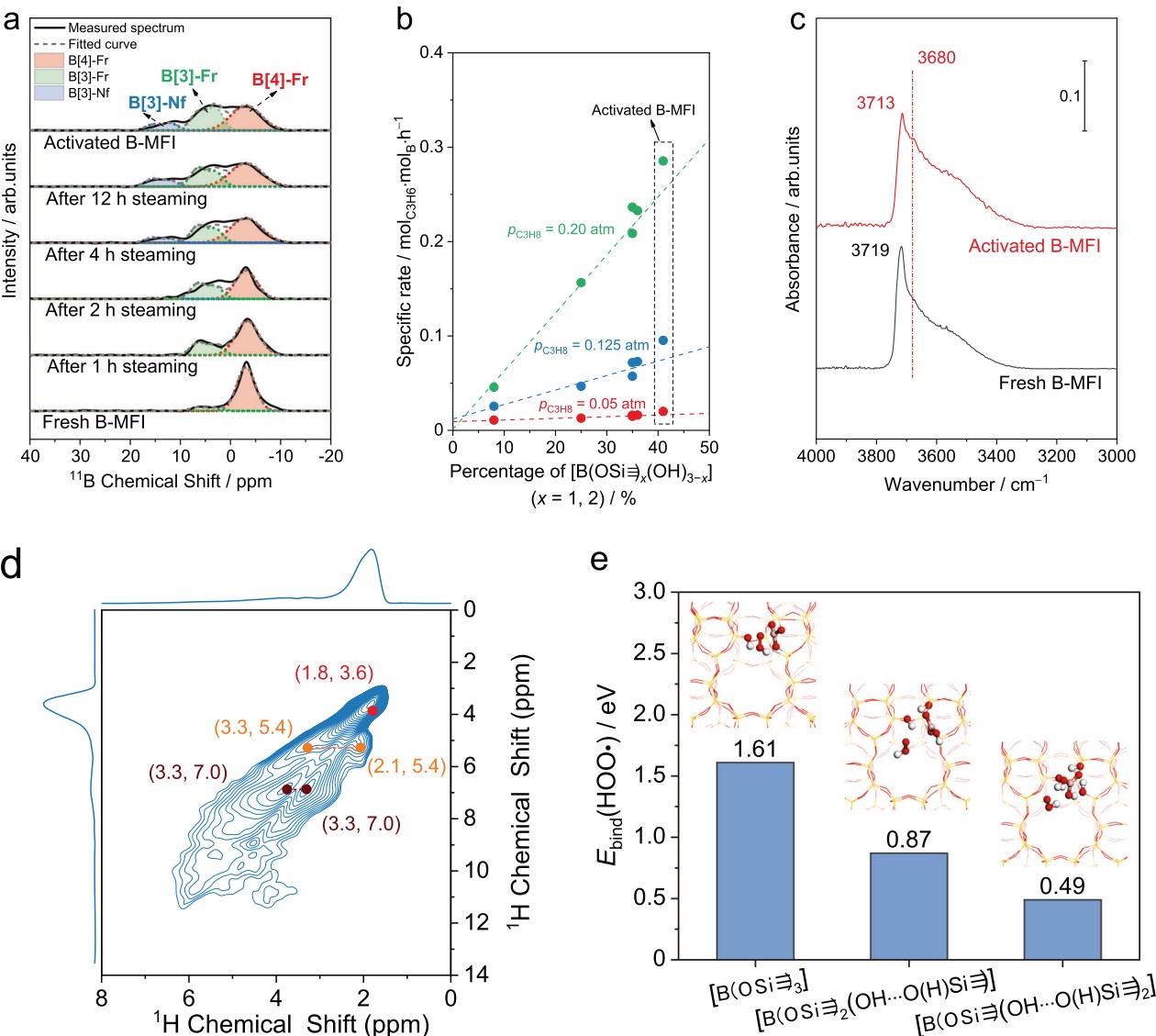

**Fig. 4 | Transformation of boron species in B-MFI during steaming treatment and its impact on ODHP mechanism. a** $^{11}$B NMR spectra and peak fitting of boron species on fresh, steamed and activated B-MFI. **b** Correlation between the specific ODHP activity and the percentage of [B(OSi≡)$_{3-x}$(OH)$_x$] in B-MFI at various $p_{C3H8}$ values. **c** Transmission IR spectra of dehydrated fresh and activated B-MFI under vacuum at 823 K. **d** 2D $^1$H-$^1$H DQ MAS NMR spectrum of activated B-MFI. **e** Calcualted binding energy of HOO• on different boron sites. Balls and sticks in the inset pictures represent atoms: hydrogen (white), boron (pink), oxygen (red), and silicon (yellow).

(Fig. 4b) The slope of fitted lines in Fig. 4b, which is the growth rate of the ODHP activity with the content of [B(OSi≡)$_{3-x}$(OH)$_x$], is highly dependent on $p_{C3H8}$. This is beacuse the supra-linear kinetics on steamed and activated B-MFI substantially enhances the ODHP rate at higher $p_{C3H8}$. Due to the conversion of B[3]-Fr to B[4]-Fr induced by water uptake at room temperature[21,34], the B[4]-Fr /B[3]-Fr ratios determined based on $^{11}$B NMR spectra collected at room temperature are likely overestimated compared to those at the reaction tempera-ture, however, the increasing trend of the B[3]-Fr/B[4]-Fr ratio with the duration of the steaming treatment should hold regardless. We note that the observation that the steaming treatment itself is insuffi-cient to fully activate B-MFI has been made on other boron-based catalysts in ODPH[50]. For example, the steaming treatment can only reduce the duration of, rather than replace, the induction period in ODHP (Fig. S9). Since the key difference between the steaming treat-ment and ODHP is the presence of reactive intermediates in the latter, we speculate that radical species generated in ODHP could help acti-vate boron-based catalysts, though the exact mechanism remains unclear.

Hydroxylation of isolated boron sites during steaming is further supported by IR results. (Fig. 4c, S16 and S17) The intensity of the band attributable to intraporous B−OH groups with hydrogen bond inter-action (~3680 cm$^{-1}$)[21,51,52] on activated B-MFI is stronger than that of fresh B-MFI at reaction temperature (823 K, Fig. 4c), which is strong evidence supporting the formation of B−OH group during the steam-ing treatment and the induction period. The peak located at 3710-3720 cm$^{-1}$ is typically attributed to the silanol group on the intraporous and external surface of zeolite crystals[21,51,52]. Transmission IR spectra collected at different temperatures confirm that B[3]-Fr instead of B[4]-Fr is also the dominant boron species on activated B-MFI at ele-vated temperature, which is consistent with the discussion of fresh B-MFI. (Figs. S16 and S17) Combined with $^{11}$B NMR results, the appearance of B−OH group indicates that the isolated boron sites are derived from the hydroxylation of [B(OSi≡)$_3$] to [B(OSi≡)$_{3-x}$(OH)$_x$]. (Fig. 4c and S18)

Results of $^1$H NMR experiments on activated B-MFI indicate the spatial proximity between boron hydroxyl in [B(OSi≡)$_{3-x}$(OH)$_x$] and silanol group. In the $^1$H MAS NMR spectrum of activated B-MFI (Fig. S19),

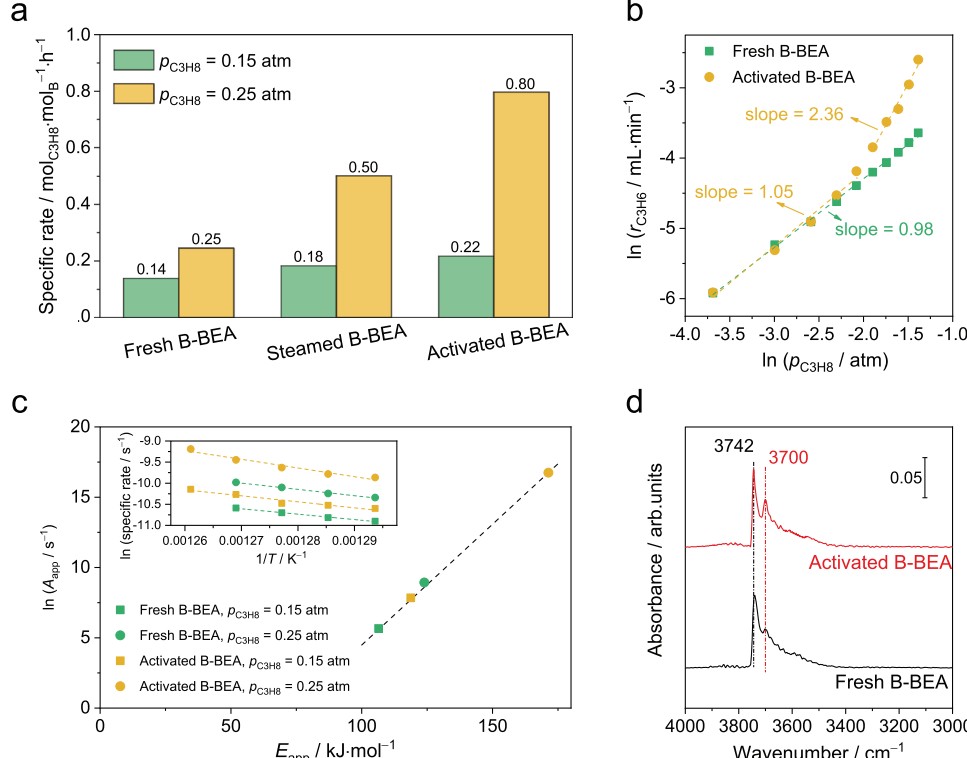

**Fig. 5 | Influence of steaming on ODHP properties and hydroxylation of B-BEA.**
**a** Specific $C_3H_6$ formation rate of different B-BEA samples. Reaction conditions: 803 K, total gas flow = 40 mL·min⁻¹, 50 mg catalyst, $p_{O_2}$ = 0.125 atm with balancing $N_2$. **b** The reaction order of $r_{C_3H_6}$ with respect to $p_{C_3H_8}$ on fresh and activated B-BEA samples. Reaction conditions: 773 K, total gas flow = 40 mL·min⁻¹, $p_{O_2}$ = 0.125 atm with balancing $N_2$. **c** The compensation effect between $E_{app}$ and $A_{app}$. The inset are corresponding Arrhenius plots. Reaction conditions: 773 to 793 K, total gas flow = 40 mL·min⁻¹, 50 mg catalyst, $p_{O_2}$ = 0.125 atm with balancing $N_2$. **d** Transmission IR spectra of dehydrated fresh and activated B-BEA in the OH stretching vibration region under vacuum at 373 K.

$^1H$ signals at 1.8, 2.1 and 3.1 ppm were observed, which could be assigned to terminal silanol groups (≡SiOH), silanol groups adjacent to oxygen (≡SiOH···O(H)X, X = Si≡ or B≡, "···" represents hydrogen bonding), and boron hydroxyl group adjacent to oxygen (=BOH···O(H)X, X = Si≡ or B≡)[21]. $^1H$-$^{11}B$ rotational-echo double resonance (REDOR) experiment was then conducted to identify which hydrogen species are adjacent to boron. (Fig. S20) Besides [≡SiOH···O(H)X] at 2.1 ppm and [=BOH···O(H)X] at 3.1 ppm, a $^1H$ signal at 2.5 ppm was observed in $^1H$-$^{11}B$ REDOR experiment, which is assigned to silanol hydroxyl groups adjacent to boron [≡SiOH···B≡] according to previous studies[21,29]. Since most boron species are still in the zeolite framework (Fig. 4a), the presence of silanol adjacent to boron supports the existence of [B(OSi≡)$_{3-x}$(OH)$_x$] in activated B-MFI. 2D $^1H$-$^1H$ double-quantum (DQ) MAS NMR spectra are shown in Fig. 4d. The autocorrelation signal at (1.8, 3.6) ppm is derived from the interaction between terminal silanol hydroxyl groups[29]. The presence of the off-diagonal peak pair at (3.3, 5.4) and (2.1, 5.4) ppm indicates the spatial proximity of the silanol and boron hydroxyl groups [=BOH···O(H)Si≡] ($\delta_{DQ}$(5.4 ppm) = 2.1 ppm + 3.3 ppm), supporting the hydroxylation of B−O−Si linkage during the activation of B-MFI. The peak pair at (3.7, 7.0) and (3.3, 7.0) ppm is likely derived from the spatial proximity of boron hydroxyl groups. Since the boron content is B-MFI is relatively low (the molar ratio of Si/B ≈ 60), it is unlikely for two framework boron atoms to be located close to one another. Thus, this NMR spectral feature could be better attributed to [B(OSi≡)(OH)$_2$], where two hydroxyl groups are bonded to the same boron atom. However, the existence of [B(OSi≡)$_2$(OH)] species cannot be excluded in activated B-MFI by these experiments. The strong signals at 2.1 and 2.5 ppm in the $^1H$-$^{11}B$ REDOR spectra of activated B-MFI suggest that most boron hydroxyl groups interact extensively with adjacent silanol groups (Fig. S20), and thus the [B(OSi≡)$_{3-x}$(OH)$_x$] species could be more accurately denoted by [B(OSi≡)$_{3-x}$(OH···O(H)Si≡)$_x$] to highlight to spatial proximity of boron and silanol groups.

DFT calculations suggest that the hydroxylation of isolated boron sites is beneficial to the stabilization of potential gas-phase radicals. (Fig. 4e) Hydroperoxyl radical (HOO•) was proposed as a potential H-abstractor of alkane in oxidative dehydrogenation[11,12]. Our previous study show that treating HOO• as the active intermediate could rationalize the observed second-order dependence on $p_{C_3H_8}$ on boron-based catalysts[12]. The binding energy of HOO• on [B(OSi≡)$_3$] was calculated to be 1.61 eV (more positive values indicate less favorable binding), which gradually decreased to 0.87 and 0.49 eV as one and two B−O−Si bonds were hydrolyzed, respectively (Fig. 4e). Stabilization of HOO• on hydroxylated framework boron species could be largely attributed to the hydrogen-bonding interaction between HOO• and the hydroxyl group(s).

## ODHP on B-BEA

To probe the generality of findings on B-MFI, effect of steaming treatment on ODHP activity and kinetics on B-BEA was investigated. XRD and $N_2$ physisorption results confirm the framework type and crystallinity of the synthesized B-BEA. (Figs. S21, S22 and Table S8) $^{11}B$ NMR spectra indicate that most boron atoms are incorporated into the zeolite framework and remain unchanged during the steaming treatment. (Figs. S23, S24 and Table S9) After 12 h of steaming treatment, the ODHP activity of fresh B-BEA increases from 0.25 to 0.50 mol$_{C_3H_6}$·mol$_B^{-1}$·h⁻¹ at $p_{C_3H_8}$ = 0.25 atm. (Fig. 5a) Similar to B-MFI, fully steamed B-BEA can be further activated in ODHP during the induction period (Fig. S25). The apparent $C_3H_8$ order of B-BEA also increases from 0.98 to 2.36 as $p_{C_3H_8}$ increases beyond 0.15 atm. (Fig. 5b) The measured $E_{app}$ and ln($A_{app}$) also exhibits the linear correlation similar

to that on B-MFI. (Fig. 5c and Table S10) Overall, the catalytic behaviors of B-BEA and B-MFI in ODHP are consistent.

Hydroxylation of boron sites also occurs during the steaming treatment on B-BEA. The conversion from B[4]-Fr to B[3]-Fr, which corresponds to the hydroxylation of isolated boron site, was also observed on activated B-BEA in $^{11}$B NMR spectra[34]. (Fig. S23 and Table S9) A shoulder peak at −103 ppm for [Si(OSi≡)$_3$(OH)] in $^{29}$Si NMR spectra appears after steaming treatment, implying the hydrolysis of Si−O−X[53]. (Fig. S24) The increase of B−OH is further supported by IR spectra. (Fig. 5d) The IR peaks of isolated Si−OH (3742 cm$^{-1}$) and isolated B−OH (3700 cm$^{-1}$) were observed in OH stretching vibration region on B-BEA[22,27,53]. The intensity of IR peak corresponding to B−OH becomes much stronger on activated B-BEA than that on fresh B-BEA, indicating the hydrolysis of B−O−Si linkages. The similarities in the correlation between boron speciation and ODHP kinetics on B-MFI and B-BEA suggest that effect of the degree of framework boron hydroxylation on ODHP is general among isolated boron sites embedded in zeolites.

## Discussion

### Correlation among kinetic variables of ODHP on boron-based catalysts

Trends of $C_3H_8$ reaction order and $E_{app}$ provide insights into ODHP mechanisms on isolated boron sites in zeolite frameworks. There is a general linear correlation between $C_3H_8$ reaction order and $E_{app}$ on boron-containing catalysts in ODHP (Fig. 6a). Our recent work demonstrated that this correlation among aggregated boron species stemmed from the gradual shift from a surface-mediated mechanism to a gas-phase radical mechanism as the $C_3H_8$ reaction order increased beyond unity (grey symbols in Fig. 6a)[27]. Surface-mediated mechanisms typically lead to first-order kinetics with respect to $C_3H_8$, while gas-phase radical mechanisms could lead to $C_3H_8$ reaction orders of up to 3[27]. It should be noted that kinetic studies only probe the most populated reaction pathway, so that a first-order kinetics does not rule out the presence of radicals in the reaction system. Instead, the radical-mediated pathway is insufficiently populated to contribute significantly to the overall reactivity, potentially due to the slow production rates or fast quenching of radicals. Since B-MFI and B-BEA fit in the same trend (blue and red symbols in Fig. 6a), we propose that ODHP on isolated boron sites also have contributions from both the surface-mediated and gas-phase radical mechanisms, but with a key difference. Aggregated boron species, e.g., $B_2O_3$ and h-BN, were known to mediate gas-phase radical pathway, so the first-order kinetics for $C_3H_8$ on $B_2O_3$ nanoparticles in mesopores of SBA-15 and MCM-41 at low $p_{C3H8}$ (< 0.10 atm) was attributed to the rapid quenching of radicals by colliding with the pore walls[27]. On fresh B-MFI and B-BEA, first-order

kinetics for $C_3H_8$ was observed within the entire $p_{C3H8}$ range investigated (0.025 atm < $p_{C3H8}$ < 0.25 atm), suggesting that [B(OSi≡)$_3$] likely only facilitates the surface-mediated pathway in ODHP. The pore sizes of B-MFI (0.55 nm) and B-BEA (0.67 nm) are smaller than those in MCM-41 (3.2 nm) and SBA-15 (8.7 nm), so quenching of gas-phase radicals is expected to be more effective in zeolites. Meanwhile, the difference in pore sizes among these materials is minor in comparison to the mean-free-path of gas at ODHP temperature (~150 nm)[27]. In addition, second-order kinetics for $C_3H_8$ was observed on activated B-MFI and B-BEA, confirming that the size of zeolitic pores alone cannot suppress the gas-phase radical pathway at high $p_{C3H8}$. The degree of its hydroxylation of framework boron species plays a decisive role, i.e., [B(OSi≡)$_3$] for the surface-mediated pathway and [B(OSi≡)$_{3-x}$(OH⋯O(H)Si≡)$_x$] for the gas-phase radical pathway in ODHP. (Fig. 4e) The detection of hydroxyl groups on spent h-BN and $B_2O_3$ catalysts, on which the radical mechanism is dominant in ODHP, is also consistent this hypothesis[16–18].

The highly consistent linear correlation between activation enthalpies and entropies in ODHP on B-MFI at different stages of activation (Fig. 6b and S26) indicating that the ODHP mechanism is largely independent of zeolite framework type but sensitive to the degree of hydroxylation or the coordination environment of boron. As the extent of hydroxylation in B-MFI increases from 8 to 41% (Fig. 6b), both $\Delta H^{\ddagger}$ and $\Delta S^{\ddagger}$ increases, suggesting increasing contributions from the gas-phase radical mechanism to the overall reactivity. The activation enthalpy-entropy correlation on B-MFI could be rationalized as changes in the relative contributions from the surface-mediated and gas-phase radical mechanisms with varying [B(OSi≡)$_3$]/[B(OSi≡)$_{3-x}$(OH⋯O(H)Si≡)$_x$] ratios. While framework [B(OSi≡)$_3$] is unique in being able to facilitate ODHP only through a mostly surface-mediated pathway, hydroxylated boron species, regardless whether in isolated or aggregated form, are able to enable both mechanisms, with the gas-phase radical mechanism being dominant at high $p_{C3H8}$. Direct comparison of ODHP between boron-containing zeolites and aggregated boron species, e.g., $B_2O_3$ and h-BN, is hampered by the difficulty in determining the density of surface boron sites. When normalized by the mass loading of boron, supported $B_2O_3$/SBA-15 catalyst with a low boron loading (0.5 wt%, $B_2O_3$ basis) exhibits comparable ODHP activity with borosilicate zeolites (Fig. S27). The specific ODHP activity of h-BN is much lower, likely due to the much lower dispersion of boron. With the concomitant evolution of the degree of hydroxylation of framework boron species and kinetic variables, including $C_3H_8$ reaction order, $E_{app}$ and $A_{app}$ on B-MFI, it could be inferred that [B(OSi≡)$_3$] is likely responsible for the surface-mediated mechanism, while [B(OSi≡)$_{3-x}$(OH⋯O(H)Si≡)$_x$] stabilizes key radcial species (such as HOO•) to initiate the gas-phase radical pathway, leading to a second-order kinetics with respect to $C_3H_8$ (Fig. 7).

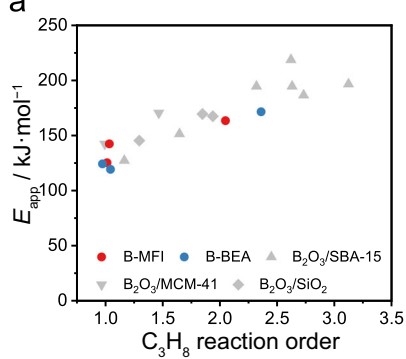
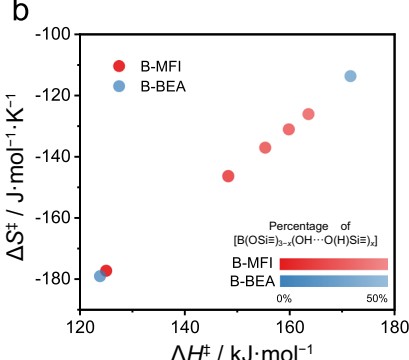

**Fig. 6 | Correlation among kinetic variables in boron-catalyzing ODHP. a** Correlation between apparent $C_3H_8$ order and $E_{app}$ on different B-containing catalysts in ODHP. The data of supported $B_2O_3$ catalysts (grey symbols) comes from our previous study[27]. **b** The influence of the degree of hydroxylation on the compensation effect of activation enthalpies and entropies in ODHP on B-MFI. The color of dots represent the percentage of [B(OSi≡)$_{3-x}$(OH⋯O(H)Si≡)$_x$] in borosilicate zeolites, quantified by the results of $^{11}$B NMR spectra. (Tables S7, S9).

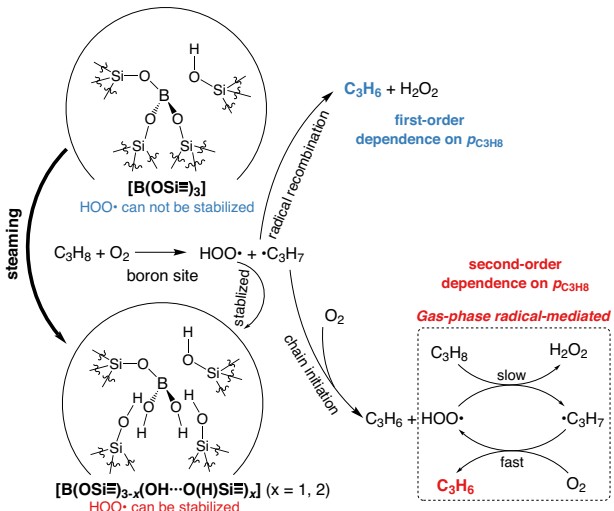

**Fig. 7 | Schematic illustration of boron species transformation on B-MFI and ODHP mechanisms via different pathways.** ODHP reaction pathways and $C_3H_8$ reaction order on different isolated boron sites in B-MFI are presented. $C_3H_6$ formation via surface-mediated pathway on $[B(OSi\equiv)_3]$ is shown in the upper part and gas-phase radical pathway on $[B(OSi\equiv)_{3-x}(OH\cdots O(H)Si\equiv)_x]$ is shown in the lower part.

In summary, we demonstrated that the degree of hydroxylation of isolated boron sites in zeolites has a major impact on the rate and mechanism in ODHP. Steaming treatment markedly promotes the ODHP activity of B-MFI by hydrolyzing framework $[B(OSi\equiv)_3]$ to $[B(OSi\equiv)_{3-x}(OH\cdots O(H)Si\equiv)_x]$. The hydroxylated boron sites are able to facilitate the gas-phase radical reaction by stabilizing radical species, which leads to more than an order of magnitude increase in the $C_3H_6$ formation rate on B-MFI. This mechanism is supported by the shift from first- to second-order kinetics with respect to $C_3H_8$ as $[B(OSi\equiv)_3]$ is converted to $[B(OSi\equiv)_{3-x}(OH\cdots O(H)Si\equiv)_x]$. Similar observations were made on B-BEA, suggesting the generality of proposed mechanism. The linear correlation between the activation enthalpy and entropy on B-MFI with different degrees of hydroxylation shows that the relative contributions of the surface-mediated and the gas-phase radical-mediated mechanisms determine the measured kinetic variables.

## Methods

### Catalysts preparation

**Materials.** Boric acid ($H_3BO_3$), $25\,wt\%$ tetrapropylammonium hydroxide (TPAOH) and $25\,wt\%$ tetraethylammonium hydroxide (TEAOH) were purchased from Alfa Aesar. Amorphous silica ($SiO_2$), sodium hydroxide (NaOH) and sodium metaborate tetrahydrate ($NaBO_2\cdot4H_2O$) were purchased from Aladdin Industrial Inc. Silicalite-1 and H-ZSM-5 samples were purchased from Tianjin Nankai University Catalyst Co., Ltd. Home-made secondary distilled water with conductivity less than $1.5\,\mu S\cdot cm^{-1}$ was used in preparation.

**Synthesis of B-MFI zeolite.** B-MFI zeolites were prepared by a solvent-free method according to the previous procedure[21]. In a typical synthesis, 0.03 g of $H_3BO_3$ and 2 g of $SiO_2$ were firstly mixed in an agate mortar. Then, 2.6 g of $25\,wt\%$ TPAOH solution was added and the mixture was ground for 15 min. The mixture was transferred into a 10 mL Teflon autoclave and heated at 453 K for 72 h. After the thermal treatment, the obtained white powders were dried at 373 K for 12 h and finally calcined at 873 K for 4 h in flowing air. The as-prepared sample is referred to as fresh B-MFI. The steaming treatment was carried out at 823 K and $p_{H2O} = 1.8\,kPa$ under air flow ($40\,mL\cdot min^{-1}$). Activated B-MFI refers to fresh or steamed B-MFI sample which was treated under ODHP condition (803 K, total gas flow = $40\,mL\cdot min^{-1}$, $p_{C3H8} = 0.25\,atm$,

$p_{O2} = 0.125\,atm$ with balancing $N_2$) for at least 6 h to finish the induction period.

**Synthesis of B-BEA zeolite.** B-BEA zeolites with *BEA framework were prepared by a hydrothermal method. In a typical synthesis, 21.5 g of $25\,wt\%$ TEAOH was mixed with 11.0 g of water with stirring. Then, 0.10 g of NaOH and 0.16 g of $NaBO_2\cdot4H_2O$ were added, following by stirring for 30 min. Then 4.80 g of $SiO_2$ was added into the solution and stirred for another 6 h. The mixed gel was transferred into a 100 mL Teflon autoclave and heated at 413 K for 6 days. After cooling down to room temperature, the obtained product was filtered, washed with water, dried at 373 K and finally calcined at 873 K for 4 h in flowing air. The as-prepared sample is referred to as fresh B-BEA. The steaming condition and the definition of activated B-BEA are same as B-MFI.

**Characterization.** X-ray diffraction (XRD) patterns were collected on a PANalytical X-Pert3 Powder equipped with a Cu $K_\alpha$ radiation ($\lambda = 1.54056\,\text{Å}$). $2\theta$ of X-ray diffraction pattern was recorded from 5° to 50° with a scan speed of 3°/min.

MAS $^{11}$B and $^{29}$Si solid-state NMR were performed on a Bruker AVANCE III 400 MHz spectrometer at resonance frequencies of 128.42 and 79.52 MHz for the $^{11}$B and $^{29}$Si nucleus, respectively, with a 2.5 mm MAS probe at a spinning rate of 20 kHz. The $^{11}$B MAS NMR spectra were collected using a single-pulse sequence with a $\pi/12$ pulse length of $3.3\,\mu s$ and a recycle delay of 3 s. The $^{29}$Si MAS NMR spectra were collected using a single-pulse sequence with a $\pi/12$ pulse length of $0.4\,\mu s$ and a recycle delay of 1 s. Before the NMR tests, the samples were dehydrated at 373 K under vacuum for 24 h. The parameters used for quantifying $^{11}$B NMR spectrum was obtained according to the reported method[54]. The estimated correction coefficient for the satellite transitions of tetra-coordinated boron $I_{cl}(ST, B[4])$ is 1.04, and the estimated correction coefficient for the central transitions caused by quadrupolar coupling of tri-coordinated boron $I_{cl}(CT, B[3])$ is 0.93. The corrected factor of measured $^{11}$B NMR area $I(B[4]/B[3])$ is 1.12.

The $^{1}$H spin-echo MAS NMR, $^{1}$H-$^{11}$B symmetry-based rotational-echo double-resonance (REDOR) MAS NMR and two-dimensional (2D) $^{1}$H-$^{1}$H DQ-SQ MAS NMR experiments were performed on a Bruker Avance III 600 spectrometer equipped with 14.1 T and 89 mm wide-bore magnet using 3.2 mm H-X-Y triple resonances MAS probe with the corresponding Larmor frequency of 600.13 MHz for $^{1}$H. Prior to the above experiments, the samples were dehydrated with a home-built vacuum line at 693 K for 12 h. The chemical shift of $^{1}$H NMR were referenced to adamantane at 1.74 ppm. The $^{1}$H spin-echo MAS NMR spectra were recorded with a $\pi/2$ pulse width of $4.0\,\mu s$, a recycle delay of 5 s, an echo time of 0.33 ms and a spinning rate of 12 kHz. The $^{1}$H-$^{11}$B REDOR spectra were performed with a spinning rate of 12 kHz and a recycle delay of 5 s which accumulated 32 scans. A $\pi/2$ pulse length of $4.0\,\mu s$ and $\pi$ pulse length of $8.0\,\mu s$ were used on the $^{1}$H channel with SR4 recoupling pulse. The recoupling time was 0.66 ms. The saturation pulse on the $^{11}$B channel was with duration of $125\,\mu s$ (1.5 Tr). The $^{1}$H-$^{1}$H DQ-SQ MAS NMR spectra were excited and reconverted with the POST-C7 pulse sequence with a spinning rate of 12 kHz and a recycle delay of 2 s. The increment in the indirect dimension ($t_1$) was set to $83.34\,\mu s$, and 32 scans were acquired for each $t_1$ increment.

Fourier transform infrared (FTIR) spectroscopy experiments were performed on a Bruker Invenio-S spectrometer in a custom transmission cell by using pyridine as the probe molecule. The samples were pressed into self-supporting wafers of about 10 mg and loaded into a sample holder for vertical alignment in the infrared beam. Spectra presented were 64 coadded scans per spectrum at a spectral resolution of $4\,cm^{-1}$. Vacuum levels of < 0.01 mTorr in the transmission cell were achieved by connecting to a vacuum manifold equipped with a mechanical pump (Agilent) and a diffusion pump (Agilent). The transmission cell was heated by heating rings controlled by a temperature programmed PID controller. Prior to all experiments, the

samples were evacuated at 823 K for 1 h to remove adsorbed molecules. The samples were then cooled down to 298 K under vacuum for further characterization.

Scanning electron microscopy (SEM) experiments were performed on a Zeiss Merlin Compact electron microscope with an acceleration voltage of 2 kV.

The element compositions of B-MFI, ZSM-5 and B-BEA zeolites were measured by inductively coupled plasma-atomic emission spectroscopy (ICP-AES) over Teledyne Leeman Labs Prodigy 7 after digestion with 2 *vol*% HF.

The N$_2$ adsorption-desorption isotherms were measured at 277 K over a Micromeritics Tristar II 3030 analyzer. Before the tests, all the materials were evacuated at 363 K for 1 h and 573 K for 6 h. Brunauer-Emmett-Teller (BET) surface area are calculated from adsorption data in range of $p/p_0 = 0.05-0.30$. The micropore evaluation was used t-plot method based on Harkins-Jura model. The linearization of t-plot is in range from thickness = 3.5–5 Å, corresponding to $p/p_0 = 0.10-0.27$.

**Computational method.** The unit cell of B-MFI is obtained from the peak positions in XRD patterns with the cell parameters of $a = 20.07$ Å, $b = 19.88$ Å, $c = 13.39$ Å, $\alpha = \beta = \gamma = 90°$. DFT calculations using periodic boundary condition (PBC) models are carried out with the Quickstep module of the CP2K (version 7.1) package[55]. The Perdew-Burke-Ernzerhof (PBE)[56] functional is used in combination with Godecker-Teter-Hutter (GTH) pseudopotentials[57] and the molecularly optimized double-$\zeta$ polarization quality Gaussian basis sets (DZVP-MOLOPT-SR-GTH)[58]. An 800 Ry energy cutoff, a 70 Ry relative energy cut-off, and 4 grid levels are employed. The binding energy of hydroperoxyl radical (HOO•) on isolated boron sites with different degree of hydroxylation was calculated as

$$E_{\text{bind}}(\text{HOO·}) = E_{\text{B-MFI-OOH}} - E_{\text{B-MFI}} - E_{O_2} - \mu_H \qquad (1)$$

where $E_{\text{B-MFI-OOH}}$, $E_{\text{B-MFI}}$, $E_{O2}$ and $\mu_H$ refer to the calculated energy of B-MFI-OOH intermediate, the isolated boron sites, O$_2$ in gas phase and the chemical potential of H which referenced from water formation $1/2 H_2 + 1/4 O_2 = 1/2 H_2O$ ($\mu_H = 1/2 E_{H_2O} - 1/4 E_{O_2}$). According to the definition of $E_{\text{bind}}(\text{HOO•})$, more positive values indicate less favorable binding between B-MFI and HOO•.

**Catalytic testing.** Catalytic properties of ODHP were evaluated in a quartz fixed-bed tubular reactor with an inner diameter of 6 mm under atmospheric pressure. In a typical experiment, 200 mg mixed sample and SiC with 20 to 40 mesh were used in tests of catalytic performance with a mass ratio of 1:3. The catalyst bed was supported by 1 cm quartz wool lug. During the steaming treatment, water vapor was introduced into the reactor by gas bubbler with the $p_{\text{H2O}} = 1.8$ kPa. The reactor effluent was analyzed online by an Agilent 8890 gas chromatograph (GC) equipped with a TCD and an FID detector. O$_2$, CO, CO$_2$ and N$_2$ were separated by 5 A column (Agilent J&W) and quantified by TCD. Hydrocarbons (C$_2$H$_6$, C$_3$H$_8$, C$_2$H$_4$, C$_3$H$_6$, CH$_4$) were separated by HP-PLOT Al$_2$O$_3$ column (Agilent J&W) and quantified by FID. The C$_3$H$_8$ conversion was determined by Eq. (2).

$$C_3H_8 \ conversion = \frac{F_{\text{C3H8,inlet}} - F_{\text{C3H8,outlet}}}{F_{\text{C3H8,inlet}}} \times 100\% \qquad (2)$$

$F$ is the volumetric flow rate under standard temperature and pressure (STP). The specific rate of C$_3$H$_6$ formation was determined by Eq. (3).

$$Specific \ rate \ of \ C_3H_6 \ formation = \frac{L_{\text{C3H6,outlet}}}{n_{\text{boron}}} \qquad (3)$$

$L$ is the molar flow rate under STP and $n_{\text{boron}}$ is the molar amount of boron in the measured catalyst, which is determined by ICP-AES.

The selectivity of product *i* was determined by Eq. (4).

$$Selectivity = \frac{n_i \times F_{i,\text{outlet}}}{3 \times (F_{\text{C3H8, in}} - F_{\text{C3H8, out}})} \times 100\% \qquad (4)$$

where *i* represents the products C$_3$H$_6$, C$_2$H$_4$, CH$_4$, CO and CO$_2$ in the effluent gas, $n_i$ is the number of carbon atoms of component *i*, and $F_i$ is the volumetric flow rate under STP. The carbon balance (Eq. (5)) in all experiments was higher than 98%,

$$Carbon \ balance = \frac{3 \times F_{\text{C3H8,outlet}} + \sum(n_i \times F_{i,\text{outlet}})}{3 \times F_{\text{C3H8, inlet}}} \times 100\% \qquad (5)$$

The deactivation of catalysts was less than 5% in all cases.

## Data availability
The authors declare that all the relevant data within this paper and its Supplementary Information file are available from the corresponding author upon a reasonable request. Source data are provided with this paper.

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

## Acknowledgements

This work is supported by the National Natural Science Foundation of China (22172001 and 22108006) and Beijing National Laboratory for Molecular Sciences. H.T. acknowledges the financial support from the Chinese Postdoctoral Science Foundation (No. 2020M680239 and 2022T150009).

## Author contributions

B.J.X. supervised the project. H.T. performed the experiment and characterization. W.Y.L. and H.X. carried out the DFT simulations and analysis. L.H.H., Y.Z.Z. and S.T.X. contributed to the characterization of borosilicate zeolites. All the authors contributed to the writing of the manuscript.

## Competing interests

The authors declare no competing interests.
