## [Peer Review File · Nature Communications]

REVIEWER COMMENTS

Reviewer #1 (Remarks to the Author):

Tian et al. studied ODHP kinetics over freshly made, steamed, and activated borosilicate zeolites. The experimental approach is comprehensive, and the findings are supported by a multitude of analytical tools. The manuscript is well-structured, and thanks to the authors' thorough approach, the experimental results are beyond doubt. They are also highly significant, as ODHP is an emerging solution for the currently limited propylene supply.

The unit of the specific rate as defined by Eq. (3) should be $\text{mol}_{\text{C3}} \cdot (\text{mol}_{\text{B}} \cdot \text{h})^{-1}$, i.e., the parentheses are missing throughout the paper.

More substantially, I somewhat disagree with two conclusions:

(1) The OOH radical is destabilized over the steamed catalyst because the computed binding energies increase. However, this is not unreasonable considering the next comment.

(2) The authors point to a growing body of literature suggesting important gas-phase contributions to ODHP. The increased apparent activation energy and reaction order are the main experimental findings supporting this. As far as the former is concerned, the authors revealed linear relationships between quantities that are, in the first approximation, unrelated, such as the pre-exponential factor and the activation energy, or the activation enthalpy and entropy. To me, this confirms the mechanism change, as proposed by the authors, but not necessarily a shift towards a gaseous process. There may also be a shift in the surface chemistry that could also explain the connection between these observables. The increase in the reaction order strongly suggests a shift from a surface-confined to a gaseous mechanism. However, the superb selectivities reported are irreconcilable with a radical-driven gaseous process. Saturated and unsaturated oxygenates, as well as carbon chain growth, are inevitable in the presence of substantial gaseous radical concentrations, neither of which is observed.

In the classical interpretation, a reaction order of two or more substantially shifts the center of gravity of the process to the gas phase. In this sense, the authors' argument is mainstream, yet I doubt its validity.

Connecting to (1), a less strongly bound OOH on the surface may be more reactive and could, perhaps, also explain at least part of the mechanism change upon steaming/activation.

I think this is a very important work that will, as the authors indeed suggest, strongly influence ODHP research. I look forward to their response to these comments and think that the catalysis community would profit greatly from the publication of this work.

Reviewer #2 (Remarks to the Author):

In this manuscript the authors capture rate data for propane ODH at various times-on-stream and with different steaming durations. They find interesting correlations between the degree of B hydroxylation and propane activation rate. They also identify a shift in the kinetic regime from first-order in propane to second-order in propane after activation (and only at higher propane concentrations). They hypothesize that these findings can be explained from a shift from surface-mediated reactions to gas-mediated reaction, and support this assertion with DFT calculations showing radical stabilization on hydroxylated B surfaces.

I find the authors' work to be very interesting, but requiring further evidence:

1. To what degree can the rate data for pre-activation catalysts (Fig 2a) be attributed to activation of propane on silanol defects, etc. on the zeolite surface? It's not immediately clear that propane is activating on framework boron sites. For example, see [10.1038/s41467-022-34967-2](https://doi.org/10.1038/s41467-022-34967-2) for an example of propane ODH on silica powder and [10.1002/anie.201914696](https://doi.org/10.1002/anie.201914696) for an example of framework B not exhibiting any activity. Do DFT calculations support a mechanism for surface-mediated propane activation prior to hydroxylation that have a lower activation energy? I do not see any rate-data reported for the as-synthesized silicalite-1 control catalyst (though I could have missed it). Was this run and if so what selectivity vs. conversion distribution is observed, if any? Further verification of B-mediated catalysis would be necessary to strongly defend the claim that initial rates prior to activation can be attributed to boron-catalyzed surface reactions. Perhaps a control experiment with a higher-loading B-zeolite or B/SiO₂ catalyst would be beneficial for generating higher conversion prior to activation.

2. Further attention should be paid to the selectivity vs. conversion distribution of the pre-activation B-containing zeolites (Fig 2b). The increasing selectivity to propylene with increasing conversion is very surprising- does this trend hold beyond 1% propane conversion?

3. Has the order in oxygen been investigated for pre-activation boron catalysts? The ~ 0.5 order in O_2 is implicated in the gas-phase chemistry, and a difference here might lend credence to a difference in mechanism prior to activation.

4. The discussion surrounding pore size could use further refinement. As the authors point out, the small pores of the pre-activation B-zeolites would be even more likely to quench radicals, possibly explaining the first-order dependence in propane over all tested concentrations prior to activation. Therefore, it's not immediately obvious that the initial zeolites exhibit solely a surface-mediated propane activation rather than rapid quenching of radical species (recalling the observation that the rate of reaction is approximately 1/10 of the activated version, meaning there are many less radicals that can escape the pores). In other words: does B only act via a gas-propagated mechanism, and only in conditions favorable to minimal radical production (low propane concentration, low rate of radical production / pre-activation B) we observe first-order kinetics due to radical quenching? Perhaps a B_2O_3/SiO_2 catalyst could be used as a control to test this hypothesis (we would expect to observe first-order kinetics even with high propane concentration prior to activation, if only hydroxylated boron demonstrates second-order kinetics).

5. In both B-BEA and B-MFI the specific activity of the fully steamed catalyst is significantly (>30%) less than that of the fully activated catalyst. Two questions:

a. Clarification- Is the specific activity given in Fig 2a after full activation or taken right after steaming?

b. Given that the specific activity levels off with steaming duration (Fig 2a) and the B[3]-Fr levels off with steaming duration (Fig 4a), can the authors explain further where the additional activity / hydroxylation / B[3]-Fr of the fully activated sample is coming from? It is a consistent, and interesting, finding in the work that the activated sample exhibits further hydroxylation than what steaming alone can provide.

6. The compensation plot (Fig 6b) and subsequent discussion could use further explanation- which data points are used to produce these plots? Are they initial rates after steaming for different durations? Could an additional x-axis of percentage of $B(OSi)_x(OH)_{3-x}$ be added to clarify the message?

I think this is a very interesting set of findings but it does not rise to the level of Nature Communications. I might recommend publication in a strong catalysis-focused journal.

Reviewer #3 (Remarks to the Author):

In their manuscript, Tian et al. conducted a systematic investigation of the kinetic behavior of the oxidative dehydrogenation of propane (ODHP) on boron-based zeolite catalysts. They proposed that various active boron species play distinct roles in these catalysts, presenting valuable new information built upon prior research on boron-containing zeolites. The insights gained from this study offer a general mechanistic framework to better understand the kinetic behavior of the ODHP on boron-based catalysts. Overall, the organization of this manuscript is great. But, in my opinion, it is necessary to provide a more rigorous identification of the active sites and engage in a more comprehensive discussion of the underlying mechanisms. By addressing these aspects, the manuscript can further illuminate the mechanistic details of boron species in boron-based zeolite catalysts. Resolving these confusions would significantly contribute to advancing our knowledge in this field.

1. Since the ODH reaction condition is similar to the steaming treatment, why the induction period still exists after the steaming treatment more than 4 h?
2. The induction period of BN catalysts and boron-based zeolites might be related to the formation and reconstruction of boron oxide species. Therefore, Ref. 47 cited by the authors on page 9 may not be appropriate. The phenomenon of the induction period over B-MFI catalyst prepared by authors is more similar to that over MFI zeolite nanosheets in ACS Catalysis, 2022, 12, 7368-7376.
3. The previously reported B-MWW zeolite (Angew. Chem., 2020, 59, 1-6) showed similar structures of "B(OSi)₃" and "B(OSi)₂OH" to B-MFI of this work, while the catalyst exhibited no conversion of propane and no boron restructuring during the ODH reaction. What is the intrinsic difference between B-MWW and B-MFI?
4. "B[3]-Nf" sites were observed after steaming for 2h, which are generally considered the strong solubility of species in water. How could they stabilize under steaming and ODH reaction conditions?
5. The data of 2D NMR need to be supplied to identified the structure of "B[3]-Fr" because not all [B(OSi)_{3-x}(OH)_x] species have catalytic activity (Science, 2021, 372, 76-80).
6. What is the kinetic behavior in the empty quartz tube?
7. Did the authors try to added the diluent such as SiC in the reactor to investigate the change activity for the fresh and activated catalyst. The surface and gas phase reaction will show the different results when adding the diluent (Org. Process Res. Dev., 2018, 22, 1644-1652).
8. It is partial to judge the reaction mode only by the evidence of the reaction order. More detailed reaction paths for the two boron species need to be proposed by DFT calculation or other means.
9. A schematic is suggested to be provided at the end to show the working mode of "B(OSi)₃" and "B(OSi)₂OH" species under the ODH reaction, which will more intuitively present the contribution of this work.
10. Please confirm if the x-axis is "0.0%-1.0%" in Figure 2b.

Bingjun Xu, Ph.D., Ge Li and Ning Zhao Chair Professor
College of Chemistry and Molecular Engineering, Peking University
Tel: +86-010-62754319 Email: b_xu@pku.edu.cn
Group website: <https://www.chem.pku.edu.cn/bingjunxu>

Aug. 26, 2023

Below we provide a point-by-point response to reviewers' comments along with the relevant changes in the revised manuscript.

Reviewer 1:

General Comments 1-1: “Tian et al. studied ODHP kinetics over freshly made, steamed, and activated borosilicate zeolites. The experimental approach is comprehensive, and the findings are supported by a multitude of analytical tools. The manuscript is well-structured, and thanks to the authors' thorough approach, the experimental results are beyond doubt. They are also highly significant, as ODHP is an emerging solution for the currently limited propylene supply.”

Response: We thank the reviewer for the positive assessment of our work and will address the comments raised below.

Comments 1-1: “The unit of the specific rate as defined by Eq. (3) should be $\text{mol}_{\text{C}_3\text{H}_6} / (\text{mol}_{\text{B}} \cdot \text{h})^{-1}$, i.e., the parentheses are missing throughout the paper.”

Response: We thank the reviewer for pointing out this mistake. We have corrected the unit of the specific rate defined by Eq. (3) as follow.

Action: We corrected the unit of specific rate in the following sentences and figures:

Line 1, Page 10

“The specific C_3H_6 formation rate on fully steamed B-MFI (with steaming duration > 4 h) reaches $0.67 \text{ mol}_{\text{C}_3\text{H}_6} \cdot \text{mol}_{\text{B}}^{-1} \cdot \text{h}^{-1}$ at $p_{\text{C}_3\text{H}_8} = 0.25 \text{ atm}$ and $530 \text{ }^\circ\text{C}$, which is nearly one order of magnitude higher than that of fresh B-MFI.”

Line 2, Page 11

“The specific C_3H_6 formation rate on activated B-MFI sample is $0.88 \text{ mol}_{\text{C}_3\text{H}_6} \cdot \text{mol}_{\text{B}}^{-1} \cdot \text{h}^{-1}$ at $p_{\text{C}_3\text{H}_8} = 0.25 \text{ atm}$, (Fig. 3d) which is comparable to the PDH activity on noble metals.⁴⁸”

Line 22, Page 17

“After 12 h of steaming treatment, the ODHP activity of fresh B-BEA increases from 0.25 to $0.50 \text{ mol}_{\text{C}_3\text{H}_6} \cdot \text{mol}_{\text{B}}^{-1} \cdot \text{h}^{-1}$ at $p_{\text{C}_3\text{H}_8} = 0.25 \text{ atm}$.”

Fig. 3d

Fig. 5a

Comment 1-2: “More substantially, I somewhat disagree with two conclusions:

(1) The OOH radical is destabilized over the steamed catalyst because the computed binding energies increase. However, this is not unreasonable considering the next comment.

(2) The authors point to a growing body of literature suggesting important gas-phase contributions to ODHP. The increased apparent activation energy and reaction order are the main experimental findings supporting this. As far as the former is concerned, the authors revealed linear relationships between quantities that are, in the first approximation, unrelated, such as the pre-exponential factor and the activation energy, or the activation enthalpy and entropy. To me, this confirms the mechanism change, as proposed by the authors, but not necessarily a shift towards a gaseous process. There may also be a shift in the surface chemistry that could also explain the connection between these observables. The increase in the reaction order strongly suggests a shift from a surface-confined to a gaseous mechanism. However, the superb selectivities reported are irreconcilable with a radical-driven gaseous process. Saturated and unsaturated oxygenates, as well as carbon chain growth, are inevitable in the presence of substantial gaseous radical concentrations, neither of which is observed.

In the classical interpretation, a reaction order of two or more substantially shifts the center of gravity of the process to the gas phase. In this sense, the authors' argument is mainstream, yet I doubt its validity.

Connecting to (1), a less strongly bound OOH on the surface may be more reactive and could, perhaps, also explain at least part of the mechanism change upon steaming/activation. ”

Response: We thank the reviewer for this comment. For the first question, the binding energy of OOH radical is defined as $E_{\text{bind}}(\text{HOO}\bullet) = E_{\text{B-MFI-OOH}} - E_{\text{B-MFI}} - E_{\text{O}_2} - \mu_{\text{H}}$, which means that a higher $E_{\text{bind}}(\text{HOO}\bullet)$ corresponds to a more unstable B-MFI-OOH. As shown in **Fig. 4e** (revised manuscript, Fig. 4d in the original manuscript), the $E_{\text{bind}}(\text{HOO}\bullet)$ decreases as the hydroxylation proceeds, indicating HOO• is more stable on hydroxylated boron sites. For the second question, we agree with the reviewer that most radical mechanisms would bring side reactions and poor product selectivity. There is a general consensus in the literature that that ODH reaction catalyzed by boron-based materials likely proceeds through a gas-phase radical mechanism (*Angew. Chem. Int. Ed.*, 2020, 59, 16527-16535; *Chin. J. Catal.*, 2022, 43, 2173-2182; *J. Am. Chem. Soc.*, 2023, 145, 7910-7917). In particular, Hermans and co-workers showed that the PDH activity of boron-based catalysts is dependent on the length of the catalyst bed when the amount of catalyst was kept constant (*Org. Process Res. Dev.*, 2018, 22, 1644-1652). These results offer irrefutable evidence for a gas phase radical mechanism. Multiple publications confirmed the low activity of propylene on boron-based catalysts by using propylene as the feed (*Science*, 2021, 372, 76-80; *Nat. Commun.*, 2023, 14, 73.). At the same time, we agree with the reviewer that the high alkene selectivity through radical reaction pathway in the ODH is unusual, and remains poorly understood. One possible explanation is that the formed alkene remains stable under the reaction condition, avoiding the formation of oxygenates and the occurrence of carbon-chain growth (*Science*, 2016, 354, 1570-1573; *Nat. Commun.*, 2023, 14, 73). Elucidation of the cause for the low activity of propylene on boron-based catalysts is outside the scope of this work.

Action: We added the above discussions in the revised manuscript (Line 5, Page 4).

Line 5, Page 4

“The high propylene selectivity of bulk boron-based catalysts in ODHP is likely due to the involvement of gas-phase radicals in activating propane, though the exact mechanism for suppressing the further conversion of propylene remains unclear.^{13, 14, 15”}

We added instruction in our revised manuscript to help readers understand the definition of $E_{\text{bind}}(\text{HOO}\bullet)$ (Line 11, Page 17).

Line 11, Page 17

“The binding energy of HOO• on [B(OSi≡)₃] was calculated to be 1.61 eV (more positive values indicate less favorable binding), which gradually decreased to 0.87 and 0.49 eV as one and two B-O-Si bonds were hydrolyzed, respectively (**Fig. 4e**).”

General Comments 1-2: “I think this is a very important work that will, as the authors indeed suggest, strongly influence ODHP research. I look forward to their response to these comments and think that the catalysis

community would profit greatly from the publication of this work.”

Response: We thank the reviewer for the positive assessment of our work again.

Reviewer 2:

General Comments 2-1: “In this manuscript the authors capture rate data for propane ODH at various times-on-stream and with different steaming durations. They find interesting correlations between the degree of B hydroxylation and propane activation rate. They also identify a shift in the kinetic regime from first-order in propane to second-order in propane after activation (and only at higher propane concentrations). They hypothesize that these findings can be explained from a shift from surface-mediated reactions to gas-mediated reaction, and support this assertion with DFT calculations showing radical stabilization on hydroxylated B surfaces.

I find the authors’ work to be very interesting, but requiring further evidence”

Response: We thank the reviewer for the encouraging comments. Based on your comments and useful advice, we have carefully revised the manuscript. The detailed revisions have been made in the revised manuscript with marked changes. The point-by-point responses to the questions/suggestions are shown as follows.

Comment 2-1: “To what degree can the rate data for pre-activation catalysts (Fig 2a) be attributed to activation of propane on silanol defects, etc. on the zeolite surface? It’s not immediately clear that propane is activating on framework boron sites. For example, see 10.1038/s41467-022-34967-2 for an example of propane ODH on silica powder and 10.1002/anie.201914696 for an example of framework B not exhibiting any activity. Do DFT calculations support a mechanism for surface-mediated propane activation prior to hydroxylation that have a lower activation energy? I do not see any rate-data reported for the as-synthesized silicalite-1 control catalyst (though I could have missed it). Was this run and if so what selectivity vs. conversion distribution is observed, if any? Further verification of B-mediated catalysis would be necessary to strongly defend the claim that initial rates prior to activation can be attributed to boron-catalyzed surface reactions. Perhaps a control experiment with a higher-loading B-zeolite or B/SiO₂ catalyst would be beneficial for generating higher conversion prior to activation.”

Response: We agree with the reviewer that confirming framework boron sites are active in the PDH is important. To verify the initial ODHP activity prior to activation is attributed to framework boron species, we conducted two control experiments.

Firstly, we synthesized silicalite-1 by similar preparation method as B-MFI, and tested the ODHP performance of as-synthesized silicalite-1. As shown in **Table R1**, no C₃H₆ formation was observed on silicalite-1. Only a small amount of CO and CO₂ were formed on silicalite-1. This result is in-line with previous study (*Science*, 2021, 372, 76-80), which found that silicalite-1 only displayed C₃H₈ combustion as a major reaction to form CO_x. Therefore, silicon species are unlikely to contribute to the C₃H₆ formation in our study.

Table R1. ODHP performance of silicalite-1. Reaction condition: 773 K, 500 mg catalyst, total gas flow = 40 mL·min⁻¹, $p_{\text{C}_3\text{H}_8} = 0.25$ atm, $p_{\text{O}_2} = 0.125$ atm with balancing N₂.

C ₃ H ₈ conversion / %	C ₃ H ₆ selectivity / %	CO selectivity / %	CO ₂ selectivity / %
<0.1	0	72	28

Secondly, we studied the relationship between the boron content in zeolite and ODHP activity of fresh samples. Increasing the boron content in zeolite may lead to the formation of non-framework boron species, thus we synthesized more B-MFI samples with lower B-to-Si ratios than the one employed in the original manuscript to ensure that most boron atoms are incorporated into the zeolite framework. As shown in **Fig. R1**, C₃H₆ formation rate of fresh B-MFI in ODHP exhibited a positive correlation to boron content in B-MFI, showing that the initial ODHP activity prior to activation could be attributed to framework boron species.

Fig. R1. The influence of boron content in fresh B-MFI on C_3H_6 formation rate in ODHP. Reaction conditions: 773 K, total gas flow = $40 \text{ mL}\cdot\text{min}^{-1}$, $p_{\text{C}_3\text{H}_8} = 0.25 \text{ atm}$, $p_{\text{O}_2} = 0.125 \text{ atm}$ with balancing N_2 .

We added the control experiments described above in the revised manuscript.

Action: We added **Table R1** and **Fig. R1** as **Table S4** and **Fig. S7**, and included the above discussions in our revised manuscript (Line 11, Page 8).

Line 11, Page 8

“To verify the initial ODHP activity prior to activation is attributed to framework boron species, we conducted two control experiments (**Table S4** and **Fig. S7**). Silicalite-1 showed no ODHP selectivity to C_3H_6 (**Table S4**), and C_3H_6 formation rate of fresh B-MFI in ODHP exhibited a positive correlation to boron content in B-MFI (**Fig. S7**). The control experiments confirmed that the framework boron species are the active sites responsible for the initial ODHP activity in fresh B-MFI.”

Comment 2-2: “Further attention should be paid to the selectivity vs. conversion distribution of the pre-activation B-containing zeolites (Fig 2b). The increasing selectivity to propylene with increasing conversion is very surprising- does this trend hold beyond 1% propane conversion?”

Response: We thank the reviewer for this comment. We employed lower space velocities to increase the propane conversion of fresh B-MFI and found that the trend of selectivity vs. conversion relationship remained and gradually approached to the level of activated B-MFI. (**Fig. R2**) At similar C_3H_8 conversions ($\sim 1.5\%$), C_3H_6 selectivity of fresh B-MFI (85.1%) is slightly lower than that of activated B-MFI (90.0%, **Fig. 3g**).

Fig. R2. Product selectivity of fresh B-MFI as a function of C_3H_8 conversion. Reaction conditions: 773 K, $p_{\text{C}_3\text{H}_8} = 0.25 \text{ atm}$, $p_{\text{O}_2} = 0.125 \text{ atm}$ with balancing N_2 , space velocity from 1500 to 24000 $\text{L}_{\text{C}_3\text{H}_8}\cdot\text{kg}_{\text{cat}}^{-1}\cdot\text{h}^{-1}$.

The change in product distributions along with the propane conversion is likely related with the presence of

gas-phase radical reaction pathway. According to the results of ^{11}B NMR (Fig. 4a), there were also some hydroxylated boron species in fresh B-MFI (about 8%). As space velocity was decreased, the residence time of gas-phase radicals produced on hydroxylated boron sites in catalyst bed was increased, which was beneficial for gas-phase radical reaction pathway, leading to high C_3H_6 selectivity. The promotion in gas-phase radical pathway on fresh B-MFI is likely responsible for the “surprising” trend of selectivity vs. conversion relationship on fresh B-MFI.

Action: We replaced previous Fig. 2b with Fig. R2, and revised the discussion on Fig. 2b. (Line 20, Page 8)

Line 20, Page 8

“The propylene selectivity increases almost linearly with the propane conversion when the conversion is below 1%, and then gradually levels off at conversion above 1.5% at 85%.”

Comment 2-3: “Has the order in oxygen been investigated for pre-activation boron catalysts? The ~ 0.5 order in O_2 is implicated in the gas-phase chemistry, and a difference here might lend credence to a difference in mechanism prior to activation.”

Response: We thank the reviewer for this comment. We agree with the reviewer that O_2 reaction order may provide important information for mechanism interpretation, especially in the study on oxidative coupling of methane. We measured the O_2 order of fresh and activated B-MFI. (Fig. R3) Unfortunately, fresh B-MFI exhibited similar ODHP trend on the dependence of p_{O_2} with activated B-MFI, and it is difficult to differentiate reaction pathway via O_2 reaction order in this study.

Fig. R3. C_3H_6 formation rate of fresh and activated B-MFI at different p_{O_2} . Reaction condition: 773 K, total gas flow = $40 \text{ mL} \cdot \text{min}^{-1}$, $p_{\text{C}_3\text{H}_8} = 0.25 \text{ atm}$ with balancing N_2 .

Comment 2-4: “The discussion surrounding pore size could use further refinement. As the authors point out, the small pores of the pre-activation B-zeolites would be even more likely to quench radicals, possibly explaining the first-order dependence in propane over all tested concentrations prior to activation. Therefore, it’s not immediately obvious that the initial zeolites exhibit solely a surface-mediated propane activation rather than rapid quenching of radical species (recalling the observation that the rate of reaction is approximately 1/10 of the activated version, meaning there are many less radicals that can escape the pores). In other words: does B only act via a gas-propagated mechanism, and only in conditions favorable to minimal radical production (low propane concentration, low rate of radical production / pre-activation B) we observe first-order kinetics due to radical quenching? Perhaps a $\text{B}_2\text{O}_3/\text{SiO}_2$ catalyst could be used as a control to test this hypothesis (we would expect to observe first-order kinetics even with high propane concentration prior to activation, if only hydroxylated boron demonstrates second-order kinetics).”

Response: We agree with the reviewer that the observed first-order kinetics with respect to propane suggests a surface-mediated mechanism but does not rule out rapid quenching of radicals due to the pore confinement. We studied the influence of surrounding pore size on the catalytic kinetics of boron sites in a recently published paper (Catal. Today, 2023, 420, 114048). At low partial pressure of propane below 0.1 atm on B_2O_3 supported on fumed silica (f- SiO_2), a close to first-order kinetics with respect to propane was observed and a second-order at higher

$p_{C_3H_8}$ at 1 wt% B_2O_3 loading. At a higher B_2O_3 loading (20 wt%), only second reaction order for propane was observed. Further, the first-order kinetics was also observed boron oxide supported in the pore of SBA-15 and attributed to the rapid quenching of radicals. These results are consistent with reviewer's hypothesis that the second-order kinetics prevails when sufficient amounts radicals are produced (higher boron loading) and minimal radical quenching occurs (unconfined). Overall, we agree with the reviewer that the conditions favorable to minimal radical production instead of a solely surface-mediated mechanism is more accurate in describing the origin of the first-order kinetics. We accepted this important suggestion of the reviewer and revised the discussions on the understanding of first-order kinetics in our manuscript.

Action: We corrected the understanding of first-order kinetics based on the above discussions in the revised manuscript (Line 19, Page 19).

Line 19, Page 19

“It should be noted that kinetic studies only probe the most populated reaction pathway, so that a first-order kinetics does not rule out the presence of radicals in the reaction system. Instead, the radical-mediated pathway is insufficiently populated to contribute significantly to the overall reactivity, potentially due to the slow production rates or efficient quenching of radicals.”

Comment 2-5: “In both B-BEA and B-MFI the specific activity of the fully steamed catalyst is significantly (>30%) less than that of the fully activated catalyst. Two questions:

- Clarification- Is the specific activity given in Fig 2a after full activation or taken right after steaming?
- Given that the specific activity levels off with steaming duration (Fig 2a) and the B[3]-Fr levels off with steaming duration (Fig 4a), can the authors explain further where the additional activity / hydroxylation / B[3]-Fr of the fully activated sample is coming from? It is a consistent, and interesting, finding in the work that the activated sample exhibits further hydroxylation than what steaming alone can provide.”

Response: We thank the reviewer for this comment. Firstly, the specific activity given in Fig. 3a (trend of activity level with steaming time was presented in Fig. 3a in the manuscript) was measured after steaming without fully activated.

For question b, there is a general linear correlation between the specific activity and fraction of B[3]-Fr species (Fig. 4b). The fraction of B[3]-Fr species in the activated sample is 41% higher than that in the fully steamed sample, and the former also exhibits a higher propylene production rate (indicated in Fig. 4b). Thus, the higher activity of the activated sample could be attributed to the generation of additional hydroxylated B[3]-Fr species that could not be formed in the steaming treatment.

Aside from B-MFI, most other boron-based catalysts, e.g., h-BN and B_2O_3 , also exhibit an induction period in first several hours in oxidative dehydrogenation. (*Science*, 2016, 354, 1570-1573; *J. Catal.*, 2018, 365, 14-23; *ACS Catal.*, 2022, 12, 7368-7376) In the case of h-BN, steaming treatment could not increase ODHP activity of h-BN, but shortens the duration of induction period. (Fig. R4) This result indicates that water is able to partially activate h-BN but cannot fully hydroxylate boron sites in h-BN in the absence of radicals generated in oxidative dehydrogenation. Thus, we propose that the gaps of specific activity and hydroxylation between steamed and activated B-MFI could be attributed to the influence of radicals generated in the ODHP reaction.

Fig. R4. The influence of steaming on the induction period of h-BN. Reaction condition: 823 K, 100 mg h-BN, total gas flow = 40 mL·min⁻¹, $p_{\text{C}_3\text{H}_8} = 0.25$ atm, $p_{\text{O}_2} = 0.125$ atm with balancing N₂. Steamed h-BN were treated under water vapor at 823 K for 6 h before test, and activated h-BN has experienced induction period.

Action: We added the above discussions on the activation and induction period of B-MFI in the revised manuscript (Line 6, Page 10 and Line 13, Page 14).

Line 6, Page 10

“The fact that activated B-MFI exhibits superior ODHP activity than fully steamed sample indicates that species involved in the reaction are able to induce structural changes at boron site that are inaccessible via steaming.”

Line 13, Page 14

“We note that the observation that the steaming treatment itself is insufficient to fully activate B-MFI has been made on other boron-based catalysts in ODHP.⁵⁰ For example, the steaming treatment can only reduce the duration of, rather than replace, the induction period in ODHP (**Fig. S9**). Since the key difference between the steaming treatment and ODHP is the presence of reactive intermediates in the latter, we speculate that radical species generated in ODHP could help activate boron-based catalysts, though the exact mechanism remains unclear.”

Comment 2-6: “The compensation plot (Fig 6b) and subsequent discussion could use further explanation- which data points are used to produce these plots? Are they initial rates after steaming for different durations? Could an additional x-axis of percentage of $\text{B}(\text{OSi})_x(\text{OH})_{3-x}$ be added to clarify the message?”

Response: We thank the reviewer for this comment. The data points in **Fig. 6b** of the original manuscript were measured by fresh and activated B-MFI under two different partial pressures of propane, and were included in **Figs. 3e** and **5c**. We agree with the reviewer that it is more meaningful to correlate the compensation plot with the degree of hydroxylation. We measured the activation enthalpy and entropy of steaming B-MFI for different durations and updated **Fig. 6b** with an additional x-axis of percentage of $[\text{B}(\text{OSi}\equiv)_{3-x}(\text{OH}\cdots\text{O}(\text{H})\text{Si})_x]$.

Action: We added the Arrhenius plots of B-MFI with different degree of hydroxylation as **Fig. S26** in revised manuscript.

Fig. S26. Arrhenius plots of B-MFI with different degree of hydroxylation. Reaction condition: 773 K to 793 K, $p_{\text{C}_3\text{H}_8} = 0.25$ atm, $p_{\text{O}_2} = 0.125$ atm with balancing N₂.

We updated **Fig. 6b** with a color bar to show the percentage of $[\text{B}(\text{OSi}\equiv)_{3-x}(\text{OH}\cdots\text{O}(\text{H})\text{Si})_x]$ in different borosilicate zeolites.

Fig. 6b. The influence of the degree of hydroxylation on the compensation effect of activation enthalpies and entropies in ODHP on B-MFI. The color of dots represent the percentage of $[B(OSi\equiv)_{3-x}(OH\cdots O(H)Si\equiv)_x]$ in borosilicate zeolites, quantified by the results of ^{11}B NMR spectra. (Tables S7 and S9)

We added the discussion on **Fig. 6b** (Line 9, Page 21).

Line 9, Page 21

“As the extent of hydroxylation in B-MFI increases from 8 to 41% (**Fig. 6b**), both ΔH^\ddagger and ΔS^\ddagger increases, suggesting increasing contributions from the gas-phase radical mechanism to the overall reactivity.”

General Comments 2-2: “I think this is a very interesting set of findings but it does not rise to the level of Nature Communications. I might recommend publication in a strong catalysis-focused journal.”

Response: We thank the reviewer for this comment, and have clarified our claims in the revised manuscript.

Reviewer 3:

General Comments 3-1: “In their manuscript, Tian et al. conducted a systematic investigation of the kinetic behavior of the oxidative dehydrogenation of propane (ODHP) on boron-based zeolite catalysts. They proposed that various active boron species play distinct roles in these catalysts, presenting valuable new information built upon prior research on boron-containing zeolites. The insights gained from this study offer a general mechanistic framework to better understand the kinetic behavior of the ODHP on boron-based catalysts. Overall, the organization of this manuscript is great. But, in my opinion, it is necessary to provide a more rigorous identification of the active sites and engage in a more comprehensive discussion of the underlying mechanisms. By addressing these aspects, the manuscript can further illuminate the mechanistic details of boron species in boron-based zeolite catalysts. Resolving these confusions would significantly contribute to advancing our knowledge in this field.”

Response: We thank the reviewer for the positive assessment of our work, and will address the concerns raised below.

Comment 3-1: “Since the ODH reaction condition is similar to the steaming treatment, why the induction period still exists after the steaming treatment more than 4 h?”

Response: We thank the reviewer for this comment. Although the conditions are similar, steaming treatment could not replace ODH reaction to achieve full activation of boron-based catalysts, which have been observed in our experiments (**Fig. R4**) and other research groups (*J. Catal.*, 2018, 365, 14-23). Considering that the biggest difference between ODH reaction and steaming is the existence of radicals, one possible explanation is that the radicals generated in ODH is beneficial for the activation of some stable boron sites, as mentioned in our response to Comment 2-5.

Comment 3-2: “The induction period of BN catalysts and boron-based zeolites might be related to the formation and reconstruction of boron oxide species. Therefore, Ref. 47 cited by the authors on page 9 may not be appropriate. The phenomenon of the induction period over B-MFI catalyst prepared by authors is more similar to that over MFI zeolite nanosheets in *ACS Catalysis*, 2022, 12, 7368-7376.”

Response: We agree with the reviewer for this comment. We have corrected the reference in the revised manuscript.

Action: We replaced reference 47 as the reviewer suggested.

Comment 3-3: “The previously reported B-MWW zeolite (*Angew. Chem.*, 2020, 59, 1-6) showed similar structures of “B(OSi)3” and “B(OSi)2OH” to B-MFI of this work, while the catalyst exhibited no conversion of propane and no boron restructuring during the ODH reaction. What is the intrinsic difference between B-MWW and B-MFI?”

Response: We thank the reviewer for this comment. This comment raises an important issue for the ODH catalyzed by isolated boron sites in zeolite and this issue has been studied by Zhou et al. (*Science*, 2021, 372, 76-80). ¹H NMR experiments in that work showed that -B[OH...O(H)-Si]₂ sites, which were derived from tetra-coordinated boron species, were absent in B-MWW. In another publication (*Angew. Chem.*, 2020, 59, 1-6), [B(OSi)₂OH] unit was detected in B-MWW, but B-MWW exhibited poor ODH activity. Based on these results, the key difference between B-MFI and B-MWW is that the tri-coordinated boron sites in B-MFI are derived from the hydrolysis of tetra-coordinated boron sites whereas the tri-coordinated boron sites in B-MWW are native (*Science*, 2021, 372, 76-80). The silanol formed with [B(OSi)₂OH] is beneficial for the generation of radicals, which was detected in the 2D ¹H-¹H DQ MAS NMR experiments in the response to Comment 3-5. The formed silanol group is beneficial to the generation of gas-phase radicals, which is supported by our DFT calculation results. (Fig. 4d in the original manuscript and Fig. 4e in the revised manuscript) It should be noted that the activity of B-MWW in ODPH remains controversial, as B-MWW prepared by Lu and co-workers showed high ODPH activity (*J. Catal.*, 2020, 385, 176-182). Based on the ¹H NMR experiments, we have revised the description of the active sites in B-MFI to make this issue clear in the manuscript. Details and revisions are given in Comment 3-5 (see below).

Comment 3-4: “B[3]-Nf” sites were observed after steaming for 2h, which are generally considered the strong solubility of species in water. How could they stabilize under steaming and ODH reaction conditions?”

Response: We agree with the reviewer that non-framework boron species are prone to loss with the presence of water. Meanwhile, the mass loading of non-framework boron species in steamed or activated B-MFI was less than 0.2 wt% based on ICP and ¹¹B NMR results. Our previous study showed that when B₂O₃ loading was lower than 5 wt%, non-framework boron species were relatively stable in the pore of SBA-15 (*Catal. Today*, 2023, 420, 114048). Therefore, the stabilization of non-framework boron species in B-MFI could be attributed to its low content and the pore of MFI-type zeolite.

Comment 3-5: “The data of 2D NMR need to be supplied to identified the structure of “B[3]-Fr” because not all [B(OSi≡)_{3-x}(OH)_x] species have catalytic activity (*Science*, 2021, 372, 76-80).”

Response: We thank the reviewer for this comment, and carried out ¹H magic angle spinning (MAS) NMR, ¹H-¹¹B rotational-echo double resonance (REDOR) and 2D ¹H-¹H double-quantum (DQ) MAS NMR experiments to further identify the structure of isolated boron sites in activated B-MFI sample. The samples were pretreated at 693 K under vacuum (pressure < 10⁻⁴ Pa) for 12 h before ¹H NMR experiments. In the ¹H MAS NMR spectrum of activated B-MFI (Fig. R5), ¹H signals at 1.8, 2.1 and 3.1 ppm are observed, which are assigned to terminal silanol hydroxyl groups (≡SiOH), silanol hydroxyl groups adjacent to oxygen (≡SiOH...O(H)X, X = Si≡ or B=), and boron hydroxyl group adjacent to oxygen (=BOH...O(H)X, X = Si≡ or B=).

Fig. R5. ^1H MAS NMR spectrum of activated B-MFI.

We further applied ^1H - ^{11}B REDOR experiment to characterize which hydrogen species are adjacent to boron. (**Fig. R6**) Besides $[\equiv\text{SiOH}\cdots\text{O}(\text{H})\text{X}]$ at 2.1 ppm and $[\text{=BOH}\cdots\text{O}(\text{H})\text{X}]$ at 3.1 ppm, a ^1H signal at 2.5 ppm is observed in ^1H - ^{11}B REDOR experiment, which is assigned to silanol hydroxyl groups adjacent to boron ($(\text{SiOH}\cdots\text{B})$), according to previous studies. (*J. Catal.*, 2014, 316, 240-250; *Science*, 2021, 372, 76-80) Since most boron species are in the framework of zeolite (^{11}B NMR spectrum, **Fig. 4a**), the presence of SiOH and BOH adjacent to boron detected by ^1H - ^{11}B REDOR supports the existence of $[\text{B}(\text{OSi}\equiv)_{3-x}(\text{OH})_x]$ in activated B-MFI.

Fig. R6. ^1H - ^{11}B REDOR spectra of activated B-MFI, combining the reference ^1H spectrum (S_0), ^1H spectrum with the application of 180° pulse trains on the ^{11}B channel (S) and the difference spectrum ($\Delta S = S_0 - S$).

2D ^1H - ^1H DQ MAS NMR spectra are shown in **Fig. R7**. The autocorrelation signal at (1.8, 3.6) ppm is derived from the interaction between terminal silanol hydroxyl groups. The off-diagonal peak pair at (3.3, 5.4) and (2.1, 5.4) ppm indicates the spatial proximity of silanol hydroxyl groups $[\equiv\text{SiOH}\cdots\text{O}(\text{H})\text{X}]$ and boron hydroxyl groups $[\text{=BOH}\cdots\text{O}(\text{H})\text{X}]$ [$\delta_{\text{DQ}}(5.4 \text{ ppm}) = 2.1 \text{ ppm} + 3.3 \text{ ppm}$]. This signal results from $[\text{=BOH}\cdots\text{O}(\text{H})\text{Si}\equiv]$ structure, verifying the hydroxylation of B-O-Si linkage in activated B-MFI. The peak pair at (3.7, 7.0) and (3.3, 7.0) ppm seems to be derived from the spatial proximity of boron hydroxyl groups, indicating $[\text{B}(\text{OSi}\equiv)(\text{OH})_2]$ species are detected in activated B-MFI. The aforementioned ^1H NMR experiments directly showed the hydroxylation of B-O-Si in activated B-MFI and detected $[\text{B}(\text{OSi}\equiv)(\text{OH})_2]$. However, we still cannot exclude the existence of $[\text{B}(\text{OSi}\equiv)_2(\text{OH})]$ species in activated B-MFI by these experiments. Considering that most boron hydroxyl groups interact with adjacent silanol groups as shown in ^1H NMR experiments, we replaced the term $[\text{B}(\text{OSi}\equiv)_{3-x}(\text{OH})_x]$ in the original manuscript with $[\text{B}(\text{OSi}\equiv)_{3-x}(\text{OH}\cdots\text{O}(\text{H})\text{Si}\equiv)_x]$ to more accurately describe the active site in activated B-MFI. We also added the results and discussions of ^1H NMR experiments in our revised manuscript.

Fig. R7. 2D ^1H - ^1H DQ MAS NMR spectrum of activated B-MFI.

Action: We added **Fig. R5**, **Fig. R6** and **Fig. R7** as **Fig. S19**, **Fig. S20** and **Fig. 4d** in the revised manuscript. We replaced $[\text{B}(\text{OSi}\equiv)_{3-x}(\text{OH})_x]$ with $[\text{B}(\text{OSi}\equiv)_{3-x}(\text{OH}\cdots\text{O}(\text{H})\text{Si}\equiv)_x]$ to describe the active site in activated B-MFI based on the results of 2D ^1H - ^1H DQ MAS NMR. We added the above discussions in the revised manuscript (Line 2, Page 16).

Line 2, Page 16:

“Results of ^1H NMR experiments on activated B-MFI indicate the spatial proximity between boron hydroxyl in $[\text{B}(\text{OSi}\equiv)_{3-x}(\text{OH})_x]$ and silanol group. In the ^1H MAS NMR spectrum of activated B-MFI (**Fig. S19**), ^1H signals at 1.8, 2.1 and 3.1 ppm were observed, which could be assigned to terminal silanol groups ($\equiv\text{SiOH}$), silanol groups adjacent to oxygen ($\equiv\text{SiOH}\cdots\text{O}(\text{H})\text{X}$, $\text{X} = \text{Si}\equiv$ or $\text{B}=\text{}$, “ \cdots ” represents hydrogen bonding), and boron hydroxyl group adjacent to oxygen ($=\text{BOH}\cdots\text{O}(\text{H})\text{X}$, $\text{X} = \text{Si}\equiv$ or $\text{B}=\text{}$).²¹ ^1H - ^{11}B rotational-echo double resonance (REDOR) experiment was then conducted to identify which hydrogen species are adjacent to boron. (**Fig. S20**) Besides $[\equiv\text{SiOH}\cdots\text{O}(\text{H})\text{X}]$ at 2.1 ppm and $[=\text{BOH}\cdots\text{O}(\text{H})\text{X}]$ at 3.1 ppm, a ^1H signal at 2.5 ppm was observed in ^1H - ^{11}B REDOR experiment, which is assigned to silanol hydroxyl groups adjacent to boron $[\equiv\text{SiOH}\cdots\text{B}\equiv]$ according to previous studies.^{21, 29} Since most boron species are still in the zeolite framework (**Fig. 4a**), the presence of silanol adjacent to boron supports the existence of $[\text{B}(\text{OSi}\equiv)_{3-x}(\text{OH})_x]$ in activated B-MFI. 2D ^1H - ^1H double-quantum (DQ) MAS NMR spectra are shown in **Fig. 4d**. The autocorrelation signal at (1.8, 3.6) ppm is derived from the interaction between terminal silanol hydroxyl groups.²⁹ The presence of the off-diagonal peak pair at (3.3, 5.4) and (2.1, 5.4) ppm indicates the spatial proximity of the silanol and boron hydroxyl groups $[\text{B}(\text{OSi}\equiv)_{3-x}(\text{OH}\cdots\text{O}(\text{H})\text{Si}\equiv)]$ ($\delta_{\text{DQ}}(5.4 \text{ ppm}) = 2.1 \text{ ppm} + 3.3 \text{ ppm}$), supporting the hydroxylation of B–O–Si linkage during the activation of B-MFI. The peak pair at (3.7, 7.0) and (3.3, 7.0) ppm is likely derived from the spatial proximity of boron hydroxyl groups. Since the boron content in B-MFI is relatively low (the molar ratio of Si/B \approx 60), it is unlikely for two framework boron atoms to be located close to one another. Thus, this NMR spectral feature could be better attributed to $[\text{B}(\text{OSi}\equiv)(\text{OH})_2]$, where two hydroxyl groups are bonded to the same boron atom. However, the existence of $[\text{B}(\text{OSi}\equiv)_2(\text{OH})]$ species cannot be excluded in activated B-MFI by these experiments. The strong signals at 2.1 and 2.5 ppm in the ^1H - ^{11}B REDOR spectra of activated B-MFI suggest that most boron hydroxyl groups interact extensively with adjacent silanol groups (**Fig. S20**), and thus the $[\text{B}(\text{OSi}\equiv)_{3-x}(\text{OH})_x]$ species could be more accurately denoted by $[\text{B}(\text{OSi}\equiv)_{3-x}(\text{OH}\cdots\text{O}(\text{H})\text{Si}\equiv)_x]$ to highlight to spatial proximity of boron and silanol groups.

”

Comment 3-6: “What is the kinetic behavior in the empty quartz tube?”

Response: We thank the reviewer for this comment. We tried to test the kinetic behavior of empty quartz tube, but no propane conversion was observed up to 823 K. Considering the reaction temperature in this study was below 803 K, the influence of empty quartz tube on ODH activity could be excluded in this study.

Comment 3-7: “Did the authors try to add the diluent such as SiC in the reactor to investigate the change

activity for the fresh and activated catalyst. The surface and gas phase reaction will show the different results when adding the diluent (*Org. Process Res. Dev.*, 2018, 22, 1644-1652).”

Response: We conducted additional experiments with SiC as the diluent to differentiate surface and gas-phase reactions in our study. As shown in **Fig. R8a**, the introduction of SiC does not influence ODHP activity of fresh B-MFI, but markedly increases C_3H_6 formation rate of activated B-MFI. According to the reference (*Org. Process Res. Dev.*, 2018, 22, 1644-1652), the increased catalytic activity induced by diluent is a characteristic of gas-phase reaction, which is owing to the extended residence time of gas-phase active species. The promoted ODHP activity of activated B-MFI with SiC dilution is linear with bed residence time. (**Fig. R8b**) These results support our hypothesis that gas-phase reaction pathway is prosperous on activated B-MFI while there is no detectable gas-phase reaction on fresh B-MFI. We have added these results in the revised manuscript.

Fig. R8. The influence of SiC diluent on ODHP activity. (a) C_3H_6 formation rate of fresh and activated B-MFI at different volumetric ratios of SiC to (SiC + B-MFI). (b) Correlation between bed residence time and C_3H_6 formation rate of activated B-MFI. Reaction condition: 793 K, $p_{C_3H_8} = 0.125$ atm, $p_{O_2} = 0.0625$ atm with balancing N_2 .

Action: We added **Fig. R8** as **Fig. S10** in the revised manuscript. We added the above discussions in the revised manuscript (Line 21, Page 11).

Line 21, Page 11

“This claim is also supported by the experiments that using SiC as a diluent to differentiate surface-mediated reactions from gas-phase radical reactions.¹⁰ The introduction of SiC does not influence ODHP activity of fresh B-MFI, but markedly increases C_3H_6 formation rate of activated B-MFI (**Fig. S10a**). Enhancements in catalytic activity induced by diluent is a characteristic of gas-phase reaction, which could be attributed to the extended residence time of gas-phase active species (**Fig. S10b**).”

Comment 3-8: “It is partial to judge the reaction mode only by the evidence of the reaction order. More detailed reaction paths for the two boron species need to be proposed by DFT calculation or other means.”

Response: We thank the reviewer for this comment. In the original manuscript, we attributed the first-order kinetics to surface-mediated reaction pathway and the supra-linear order kinetics to gas-phase radical reaction pathway. As discussed in Comment 2-4, it is more accurate to attribute the first-order kinetics to the conditions with slow radical production and/or facile radical quenching, making surface-mediated mechanism the dominant pathway in the propylene formation. As for the supra-linear kinetics, literature studies (*Org. Process Res. Dev.*, 2018, 22, 1644-1652; *Angew. Chem. Int. Ed.*, 59, 16527-16535) and our recent work (*Chin. J. Catal.*, 2022, 43, 2173-2182; *Catal. Today*, 2023, 420, 114048) have verified the nearly second-order kinetics of boron-based catalysts could be attributed to the gas-phase radical reaction pathway. Moreover, the results of adding SiC in the response to comment 3-7 also support our explanation of reaction order and reaction mode.

In the revised manuscript, we revised the explanation of the first-order kinetics as the response to Comment 2-4. We also included the experiments of adding SiC to support our viewpoints as the response to Comment 3-7.

Action: We added the discussion above in the revised manuscript (Line 19, Page 3).

Line 19, Page 3

“Hermans and co-workers found that the conversion rate of C_3H_8 was proportional to the volume of catalyst bed, rather than the catalyst loading, in the h-BN catalyzed ODHP, implying that a surface-mediated pathway is unlikely to be able to account for the majority of observed activity.¹⁰”

Comment 3-9: “schematic is suggested to be provided at the end to show the working mode of “ $B(OSi)_3$ ” and “ $B(OSi)_2OH$ ” species under the ODH reaction, which will more intuitively present the contribution of this work.”

Response: We agree with this comment, and have revised previous schematic illustration (previous Fig. 4e) of the boron species and reaction pathway based on the above modifications and moved it to the end of the revised manuscript as Fig. 7.

Fig. 7. Schematic illustration of boron species transformation on B-MFI and ODHP mechanisms via different pathways.

Action: We revised and moved previous Fig. 4e to Fig. 7 as the reviewer suggested. We added the description of the scheme in the revised manuscript (Line 22, Page 21).

Line 22, Page 21

“With the concomitant evolution of the degree of hydroxylation of framework boron species and kinetic variables, including C_3H_8 reaction order, E_{app} and A_{app} on B-MFI, it could be inferred that $[B(OSi)_3]$ is likely responsible for the surface-mediated mechanism, while $[B(OSi)_3-x(OH...O(H)Si)_x]$ stabilizes key radical species (such as $HOO\bullet$) to initiate the gas-phase radical pathway, leading to a second-order kinetics with respect to C_3H_8 (Fig. 7).”

Comment 3-10: “Please confirm if the x-axis is “0.0%-1.0%” in Figure 2b.”

Response: We thank the reviewer for this comment. In the first version of the manuscript, the x-axis in Fig. 2b is 0.0% to 1.0% due to the poor activity of fresh B-MFI. We have increased the propane conversion of fresh B-MFI to extend the x-axis in Fig. 2b as the response to comment 2-2 in the revised manuscript.

Summarizing remarks:

We would like to thank the reviewers for their insightful comments and suggestions, and we believe the resulting revisions have clarified significant points within manuscript and improved the potential reach and impact.

Thanks for your consideration,

Bingjun Xu

REVIEWERS' COMMENTS

Reviewer #1 (Remarks to the Author):

The authors have addressed all comments substantially in the review and carried out numerous experiments to obtain further evidence. Especially the one with SiC diluent has convinced me about the validity of the mechanistic proposals. I now support the publication of the revised work in Nature Communications.

Reviewer #2 (Remarks to the Author):

The authors addressed most of the reviewers' concerns. I only have two pertaining comments/questions:

Page 2: "a higher Ebind (HOO•) corresponds to a more unstable B-MFI-OOH."

This is very counter intuitive! If X binds strongly to surface Y, this should mean that Y-X is stable with respect to X and Y separately. I can't tell if higher means 'more positive' or higher in absolute value.

Page 3: "C3H6 formation rate of fresh B-MFI in ODHP exhibited a positive correlation to boron content in B-MFI, showing that the initial ODHP activity prior to activation could be attributed to framework boron species."

I do not agree with this statement.

Reviewer #3 (Remarks to the Author):

The revised manuscript is suitable for nature communications now. I have no more questions.

Bingjun Xu, Ph.D., Ge Li and Ning Zhao Chair Professor
College of Chemistry and Molecular Engineering, Peking University
Tel: +86-010-62754319 Email: b_xu@pku.edu.cn
Group website: <https://www.chem.pku.edu.cn/bingjunxu>

Sep. 24, 2023

Below we provide a point-by-point response to reviewers' comments along with the relevant changes in the revised manuscript.

Reviewer 1:

General Comments 1-1: *"The authors have addressed all comments substantially in the review and carried out numerous experiments to obtain further evidence. Especially the one with SiC diluent has convinced me about the validity of the mechanistic proposals. I now support the publication of the revised work in Nature Communications."*

Response: We thank the reviewer for the positive assessment of our work.

Reviewer 2:

General Comments 2-1: *"The authors addressed most of the reviewers' concerns. I only have two pertaining comments/questions:"*

Response: We thank the reviewer for the encouraging comments and will address the comments raised below.

Comment 2-1: *"Page 2: "a higher $E_{bind}(HOO\bullet)$ corresponds to a more unstable B-MFI-OOH." This is very counter intuitive! If X binds strongly to surface Y, this should mean that Y-X is stable with respect to X and Y separately. I can't tell if higher means 'more positive' or higher in absolute value."*

Response: We thank the reviewer for this comment. The "higher" here means "more positive". We have added necessary instructions in the section of methods to make it clear.

Action: We added necessary instructions in our revised manuscript (Line 5, Page 24).

Line 11, Page 8

Action: "According to the definition of $E_{bind}(HOO\bullet)$, more positive values indicate less favorable binding between B-MFI and $HOO\bullet$."

Comment 2-2: *"Page 3: " C_3H_6 formation rate of fresh B-MFI in ODHP exhibited a positive correlation to boron content in B-MFI, showing that the initial ODHP activity prior to activation could be attributed to framework boron species."*

I do not agree with this statement."

Response: We thank the reviewer for this comment. We agree with the reviewer that the language in our statement here could be too strong and have revised this sentence to make the statement more rigorous.

Action: We revised this statement in our revised manuscript (Line 2, Page 8).

Line 2, Page 8

“The control experiments confirmed that the framework boron species are likely to be the active sites responsible for the initial ODHP activity in fresh B-MFI.”

Reviewer 3:

General Comments 3-1: *“The revised manuscript is suitable for nature communications now. I have no more questions.”*

Response: We thank the reviewer for the positive assessment of our work.

Summarizing remarks:

We would like to thank the reviewers for their insightful comments and suggestions, and we believe the resulting revisions have clarified significant points within manuscript and improved the potential reach and impact.

Thanks for your consideration,

Bingjun Xu